# CrossWordBench: Evaluating the Reasoning Capabilities of LLMs and LVLMs with Controllable Puzzle Generation

Jixuan Leng[1], Chengsong Huang[2], Langlin Huang[2], Bill Yuchen Lin[3],
William W. Cohen[1], Haohan Wang[4], Jiaxin Huang[2]
[1]CMU, [2]WUSTL, [3]UW, [4]UIUC
jixuanl@cs.cmu.edu, jiaxinh@wustl.edu

Code: https://github.com/SeanLeng1/CrossWordBench
Dataset: https://huggingface.co/datasets/HINT-lab/CrossWordBench

## Abstract

Existing reasoning evaluation frameworks for Large Language Models (LLMs) and Large Vision-Language Models (LVLMs) predominantly assess either text-based reasoning or vision-language understanding capabilities, with limited dynamic interplay between textual and visual constraints. To address this limitation, we introduce CrossWordBench, a benchmark designed to evaluate the reasoning capabilities of both LLMs and LVLMs through the medium of crossword puzzles—a task requiring multimodal adherence to semantic constraints from **text-based clues** and intersectional constraints from **visual grid structures**. CrossWordBench leverages a controllable puzzle generation framework that produces puzzles in two formats (*text* and *image*), supports adjustable difficulty through prefill ratio control, and offers different evaluation strategies, ranging from direct puzzle solving to interactive modes suitable for agentic evaluation. Our extensive evaluation of over 20 models reveals that reasoning LLMs substantially outperform non-reasoning models by effectively leveraging crossing-letter constraints. We further demonstrate that LVLMs struggle with the task, showing a strong correlation between their puzzle-solving performance and grid-parsing accuracy. Our findings highlight the limitations of the reasoning capabilities of current LLMs and LVLMs, and provide an effective approach for creating multimodal constrained tasks for future evaluations.

## 1 Introduction

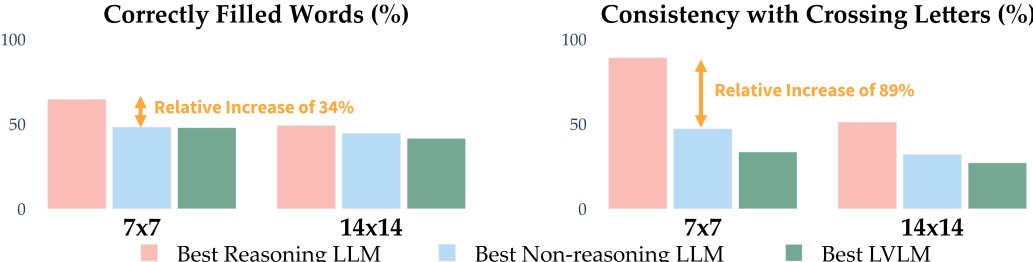

Figure 1: Performance gap between top-performing reasoning LLMs and non-reasoning LLMs/LVLMs on CrossWordBench. Reasoning LLMs achieve better overall performance and adhere more effectively to crossing-letter constraints than non-reasoning LLMs/LVLMs.

Large reasoning models (*e.g.*, OpenAI-o1 (Jaech et al., 2024) and DeepSeek-R1 (Guo et al., 2025)) have made exceptional progress on reasoning benchmarks including math problem solving (Veeraboina, 2023), coding (Austin et al., 2021) and commonsense reasoning (Clark et al., 2018). However, existing evaluation benchmarks for Large Language Models (LLMs) and Large Vision-Language Models (LVLMs) (*e.g.*, Visual Question Answering (Yue et al., 2023)) mostly focus on text-based reasoning or vision-language understanding, lacking dynamic interplay between textual and visual constraints that characterizes real-world

problem solving. Consequently, evaluating multimodal reasoning capabilities—particularly on reasoning tasks requiring both textual and visual constraints—remains challenging.

**Crossword puzzles**, a classic grid-based task in which horizontal ("Across") words and vertical ("Down") words must be filled in based on text-based clues, provide a unique testbed for such evaluations. They pose two distinct challenges: (1) question answering for each **text-based clue**, which may admit multiple correct solutions, and (2) **visual constraint satisfaction**, which requires precise letter alignment at the intersections of Across and Down entries. Prior crossword datasets (Efrat et al., 2021; Rozner et al., 2021; Kulshreshtha et al., 2022; Chen et al., 2025) often suffer from their reliance on static and copyrighted online news sources and adopt a text-centric formulation, thereby neglecting the visual structure.

In this paper, we introduce CrossWordBench, a controllable and scalable benchmark for evaluating the reasoning capabilities of both LLMs and LVLMs. CrossWordBench collects data and generates puzzles from three sources: (1) multilingual word-clue pairs from public repositories, (2) dictionary-based definitions, and (3) adapted question-answer pairs from existing benchmarks (*e.g.*, CommonsenseQA (Talmor et al., 2018)) where the answers are open-ended or unconstrained. By representing grids and clues in different formats, as shown in Figure 2, CrossWordBench facilitates the evaluation of both model types. It supports two evaluation modes: a direct puzzle-solving mode, which generates one-round responses using zero-shot Chain-of-Thought (CoT) prompts, and an interactive mode for step-by-step puzzle-solving, where grid update functions can provide intermediate visual outputs for follow-ups, thereby serving as a foundation for evaluating agents using function calling.

We evaluate over 20 state-of-the-art models, including both proprietary models (*e.g.*, GPT-4o (Hurst et al., 2024) and Claude 3.7 Sonnet (Anthropic, 2024)) and open-weight models (*e.g.*, DeepSeek-R1 (Guo et al., 2025) and Pixtral-Large-Instruct (Agrawal et al., 2024)). Our evaluation yields several notable findings: (1) **LVLMs perform significantly worse than LLMs** on CrossWordBench (as shown in Figure 1), and they struggle in OCR in vertical ("Down") word extraction. In fact, their puzzle-solving performance strongly correlates with their grid-parsing accuracy ($r = 0.94$). (2) Reasoning LLMs outperform non-reasoning models, and **benefit from both test-time scaling and increased crossing-letter constraints**. (3) Even **puzzles derived from saturated benchmarks** (*e.g.*, CommonsenseQA) **remain challenging**, highlighting the significance of structural constraints in reasoning evaluation.

## 2 Related Work

**LLMs and LVLMs Reasoning.** Recent research on the reasoning capabilities of LLMs (Ahn et al., 2024; Huang & Chang, 2022; Kojima et al., 2022; Plaat et al., 2024; Jaech et al., 2024; Guo et al., 2025; OpenAI, 2025; Huang et al., 2022) has led to the development of various approaches, including prompting-based methods (Wei et al., 2022; Yao et al., 2023; Besta et al., 2024; Chen et al., 2022) that guide LLMs through intermediate reasoning steps for solving complex problems, fine-tuning methods that train LLMs on long reasoning chains (Ye et al., 2025; Muennighoff et al., 2025; Zhao et al., 2025), and test-time scaling via self-refinement or the use of a verifier (Setlur et al., 2024; Feng et al., 2023; Wang et al., 2023; Zhang et al., 2024a; Uesato et al., 2022; Huang et al., 2025a). Moreover, a recent study from Deepseek-R1 (Guo et al., 2025) demonstrates that reinforcement learning (RL) with verifiable rewards facilitates the emergence of complex thinking processes in LLMs. Several studies have explored reasoning in the multimodal domain by constructing CoT data for fine-tuning (Xu et al., 2024) and developing verifiable problems for RL (Yang et al., 2025; Huang et al., 2025b). While these approaches have demonstrated success, existing evaluation datasets remain largely restricted to math problems (Wang et al., 2024; Lu et al., 2023; Zhang et al., 2024b).

**Crossword Puzzles in Language Model Evaluation.** Crossword puzzles have long been a focus of research in natural language processing (NLP), particularly before the advent of LLMs. Early approaches typically employed constraint satisfaction algorithms augmented by external knowledge bases. Notable examples include systems such as *Proverb* (Littman et al., 2002), *Dr. Fill* (Ginsberg, 2011), and specialized models such as *the Berkeley Crossword Solver* (Wallace et al., 2022), which incorporate a fine-tuned BERT (Devlin et al., 2019) and belief propagation. More recent studies have leveraged LLMs to address crossword puzzles

through techniques including fine-tuning (Efrat et al., 2021; Rozner et al., 2021; Sadallah et al., 2024), prompting strategies such as Tree-of-Thoughts (Yao et al., 2023), or integration with search algorithms (Saha et al., 2024), demonstrating the potential of LLMs for crosswords.

Several datasets for crossword puzzles have been proposed, covering both English (Efrat et al., 2021; Rozner et al., 2021; Kulshreshtha et al., 2022; Chen et al., 2025) and Arabic (Zeinalipour et al., 2025). However, one significant limitation of these datasets and approaches is that they rely on data from online news sources (Efrat et al., 2021; Rozner et al., 2021; Kulshreshtha et al., 2022; Chen et al., 2025) and often formulate crossword solving as a question-answering (QA) task (Sadallah et al., 2024; Efrat et al., 2021; Rozner et al., 2021; Yao et al., 2023), thereby overlooking the fundamental constraint-based nature of the problem. Moreover, all of them treat crossword solving as a text-based task, despite the inherently visual nature of crossword grids, leaving a gap in extending crossword puzzles for evaluating LVLMs. In this work, we address these limitations by introducing CrossWordBench, a framework that features controllable puzzle generation and extends evaluation to LVLMs.

# 3 Benchmark Curation

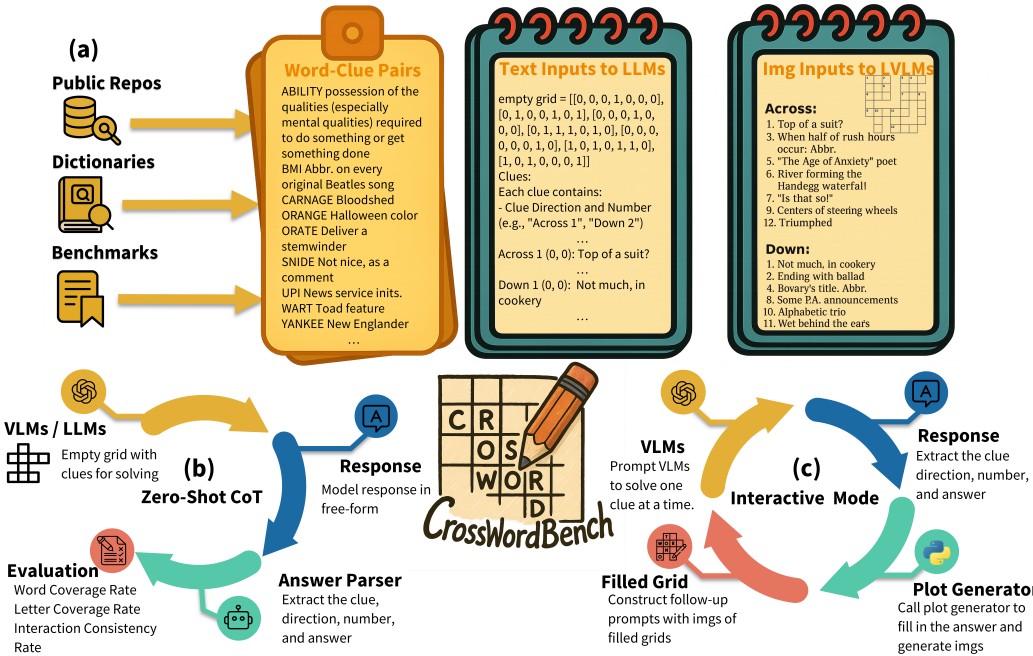

Figure 2: Framework of CrossWordBench. (a) Dataset curation process and input templates for LLMs and LVLMs; (b) Zero-shot CoT evaluation; (c) Interactive Mode Evaluation.

Online sources, including *The New York Times* and *The Los Angeles Times*, provide complete crossword puzzles. Nevertheless, directly utilizing these puzzles presents several disadvantages. **(1)** Copyright restrictions may limit their usage, and their online availability increases the probability that they have been incorporated into the pretraining datasets of contemporary models. **(2)** Online puzzles are typically static—with predefined words, themes, grid sizes, and strict formatting (e.g., black square symmetry)[1]—which not only limits their adaptability for diverse benchmark generation, but also imposes arbitrary formatting constraints that do not yield meaningful benefits for evaluation. Prior research (Mirzadeh et al., 2024) has shown that even minor modifications to questions can significantly affect model performance, particularly when performance on the original version is near saturation. **(3)** The static characteristics of these puzzles restrict the range of potential evaluation strategies. To overcome these limitations, we propose a two-stage benchmark construction strategy:

---

[1]The New York Times Crossword Requirements.
[2]A Dataset of Cryptic Crossword Clues
[3]Chinese Crosswords

**Word-Clue Pairs Curation.** We compile word-clue pairs from three source categories: **(1) Public repositories:** We selectively extract individual, often cryptic, word-clue pairs from public online repositories with samples in both English[2] and Chinese[3]. **(2) Dictionary-Based Pairs:** We collect standard English words and use their dictionary definitions from NLTK WordNet[4] as clues, referring to this category as **English Simple**. **(3) Adapted Benchmark Data:** We demonstrate that existing LLM benchmarks, which are typically designed for open-ended or multiple-choice questions without strict formatting constraints, can be transformed into word-clue pairs. In our study, we filter the CommonsenseQA (Talmor et al., 2018) **training** set for single-word answers and use the associated questions as clues.

While the possibility of data contamination may still exist for these word-clue pairs, the variations in grid design can still yield distinct crossword puzzles, and as demonstrated in Section 4.5, even when incorporating extensively tuned data such as CommonsenseQA training set, the resulting puzzles remain remarkably challenging for both LLMs and LVLMs.

Table 1: Crossword puzzle statistics for different subjects and grid sizes. Statistics are presented separately for each category, as distinct word and clue pairs are used for their construction. Additionally, the aggregated statistics for all English puzzles are included.

| Stats. | English | | Chinese | English Simple | CommonsenseQA |
|---|---|---|---|---|---|
| | 7×7 | 14×14 | 7×7 | 7×7 | 7×7 |
| Total # of puzzles | 100 | 100 | 100 | 100 | 50 |
| Total # of words | 1,193 | 3,472 | 1,327 | 1,139 | 543 |
| Unique words (% total) | 83.82% | 80.76% | 92.92% | 36.70% | 59.85% |
| Unique clues (% total) | 100% | 100% | 100% | 100% | 100% |
| **Aggregated (English 7x7 & 14x14): 200 puzzles, 74.02% unique words, 100% unique clues** | | | | | |
| *Words per Puzzle* | | | | | |
| Minimum | 11 | 22 | 11 | 11 | 9 |
| Maximum | 16 | 44 | 18 | 13 | 13 |
| Mean | 11.93 | 34.72 | 13.27 | 11.39 | 10.86 |
| *Word Length (Letters)* | | | | | |
| Minimum | 2 | 3 | 2 | 3 | 3 |
| Maximum | 5 | 12 | 5 | 5 | 5 |
| Mean | 3.59 | 4.31 | 3.02 | 3.63 | 3.77 |
| Avg blocked cells (%) | 39.51% | 45.22% | 43.37% | 39.12% | 38.37% |

**Puzzle Generation.** After collecting word-clue pairs, we implement an automatic pipeline to generate puzzles in both **text** and **image** formats, as illustrated in Figure 2. The pipeline enables (1) puzzle difficulty control by adjusting grid sizes, and (2) incremental word placement functions, which facilitate the interactive evaluation mode described in Section 4.4.

The generation framework leverages a heuristic scoring function that encourages desirable properties such as a high number of horizontal and vertical word intersections. Post-processing filters are applied to discard puzzles with low clue counts or high blocked cell ratios, thereby enhancing overall quality and structural coherence. These heuristics and filters, in combination with grid size adjustments, help control puzzle complexity. To preserve the category integrity, we ensure that clues do not overlap within each category[5].

**Puzzle Quality.** While CrossWordBench does not include human-created puzzles, its design is inspired by them. Table 1 reports overall dataset statistics—including clue count, average word length, and blocked cell ratio—which serve as proxies for puzzle quality and difficulty.

Compared human-created puzzles—for example, those used LR$^2$Bench (Chen et al., 2025), which were sourced from outlets such as *The Los Angeles Times* and *Vulture*—our generated puzzles exhibit comparable difficulty when evaluated using similar word-level metrics. Our focus on automatically generated puzzles is motivated by the need for scalable, controllable, and diverse evaluation across subjects and formats. This enables systematic variation in

---

[4]NLTK Wordnet

[5]Please check out our code for more implementation details.

difficulty through grid size, prefill ratio, and clue selection; supports step-by-step reasoning via an interaction mode; and facilitates fine-grained metric analysis beyond overall accuracy.

## 4 Experiment

In the following sections, we present an extensive empirical evaluation of CrossWordBench across multiple different model architectures and analyze their performance characteristics.

### 4.1 Experimental Setup

**Evaluation Data and Metrics.** For the main experiments, we evaluate models on the **English** set with three metrics to assess answer accuracy and adherence to crossword constraints.

- **Word Coverage Rate (WCR)**: word-level accuracy, percentage of correctly solved words.
- **Letter Coverage Rate (LCR)**: letter-level accuracy, percentage of correct letter placements.
- **Intersection Consistency Rate (ICR)**: the internal consistency of the model's answers at intersections where across and down words overlap, defined as:

$$\text{ICR} = \frac{1}{|\mathcal{I}|} \sum_{(a,d,j,k) \in \mathcal{I}} \mathbb{1}\{a[j] = d[k]\} \tag{1}$$

where $\mathcal{I}$ denotes the set of all intersections, where each tuple $(a, d, j, k)$ indicates the $j$th letter of the across word $a$ overlaps with the $k$th letter of the down word $d$. This metric reflects whether models correctly adhere to the grid structural constraints of a puzzle.

**Evaluated Models.** We evaluate a collection of proprietary and open-weight models, including both LVLMs and LLMs. For **proprietary models**, we consider state-of-the-art models such as GPT-4o (Hurst et al., 2024) and Claude 3.7 Sonnet (Anthropic, 2024) (with and without thinking mode). For **open-weight models**, our selections range from 3B to 124B parameters, such as the Qwen Series (Team, 2024; Bai et al., 2025) and Pixtral-Large-Instruct-2411 (Agrawal et al., 2024) for LVLMs. For LLMs, we include both reasoning models such as the Deepseek Series (Liu et al., 2024; Guo et al., 2025) and non-reasoning models such as Llama series (Dubey et al., 2024). The full list of models is shown in Table 2. We set decoding temperature to 0 for non-reasoning models for consistency, 0.6 for reasoning models based on commonly recommended settings, and 1.0 for certain proprietary models (e.g., used by Claude 3.7-Sonnet with Thinking). Further generation details are provided in Appendix D.3.

**Input Prompt Templates and Output Response Parsing.** For the main evaluation, we adopt the zero-shot Chain-of-Thought (CoT) (Wei et al., 2022) prompting strategy. In LVLM evaluation, the clues and the grid are both embedded within an image. In LLM evaluation, the grid is represented as a 2D binary array, with 1 indicating a blocked cell and 0 representing an unfilled cell, and is prepended to text clues in the prompt. To extract answers from responses, we leverage the structured output capabilities of o3-mini[6] to convert raw model responses into JSON format by generating dynamic Pydantic models. Detailed prompt templates and implementation details are listed in Appendix D.4 and D.2.

### 4.2 Main Results

**Reasoning LLMs substantially outperform conventional ones across metrics, with notable improvements in ICR**, as shown in Table 2. In particular, among reasoning LLMs, o3-mini achieves an ICR of 0.891 on 7x7 grids, demonstrating a strong ability to interpret and enforce grid constraints. Although DeepSeek-R1 attains the highest WCR and LCR on 7x7 grids, its ICR is slightly lower than that of o3-mini. Other reasoning models, such as QwQ-32B and R1-Distilled-Llama-70B, show moderate performance, with WCRs of approximately 0.347 and 0.387, respectively. One notable outlier is Open-Reasoner-Zero-32B, which performs poorly across all metrics, indicating that it does not effectively leverage grid constraints for reasoning. This suggests that its training—primarily focused on mathematical reasoning—does not generalize well to tasks requiring spatial and linguistic integration, thereby highlighting a key limitation in the training strategies of these reasoning models.

---

[6]https://openai.com/index/openai-o3-mini/

Table 2: Comparison of various LLMs and LVLMs on CrossWordBench **English** set across two difficulty levels using zero-shot CoT. We report the mean and standard error over 100 samples for both 7x7 and 14x14 grids. ♀ indicates that the model is a reasoning model. [†]: We use the Fireworks API for DeepSeek V3 and Llama-3.1-405B, while offical API for R1.

| Models | 7x7 | | | 14x14 | | |
|---|---|---|---|---|---|---|
| | WCR | LCR | ICR | WCR | LCR | ICR |
| *Proprietary LVLMs* | | | | | | |
| **Claude-3-7-Sonnet** | $0.479_{\pm 0.014}$ | $0.528_{\pm 0.013}$ | $0.366_{\pm 0.016}$ | $0.416_{\pm 0.009}$ | $0.449_{\pm 0.009}$ | $0.272_{\pm 0.010}$ |
| **Claude-3-7-Sonnet** ♀ | $0.365_{\pm 0.017}$ | $0.448_{\pm 0.014}$ | $0.330_{\pm 0.015}$ | $0.382_{\pm 0.009}$ | $0.428_{\pm 0.007}$ | $0.228_{\pm 0.008}$ |
| **GPT-4o-2024-11-20** | $0.348_{\pm 0.015}$ | $0.403_{\pm 0.015}$ | $0.234_{\pm 0.017}$ | $0.350_{\pm 0.009}$ | $0.393_{\pm 0.008}$ | $0.190_{\pm 0.008}$ |
| **Gemini 2.0 Pro Exp** | $0.351_{\pm 0.014}$ | $0.368_{\pm 0.014}$ | $0.339_{\pm 0.015}$ | $0.273_{\pm 0.008}$ | $0.303_{\pm 0.007}$ | $0.221_{\pm 0.007}$ |
| **Gemini 2.0 Flash** | $0.277_{\pm 0.015}$ | $0.300_{\pm 0.013}$ | $0.225_{\pm 0.013}$ | $0.260_{\pm 0.008}$ | $0.284_{\pm 0.008}$ | $0.190_{\pm 0.007}$ |
| *Open-Weight LVLMs* | | | | | | |
| **Pixtral-Large-Instruct-2411** | $0.297_{\pm 0.015}$ | $0.338_{\pm 0.014}$ | $0.198_{\pm 0.014}$ | $0.251_{\pm 0.009}$ | $0.284_{\pm 0.007}$ | $0.134_{\pm 0.007}$ |
| **InternVL2_5-78B-MPO** | $0.121_{\pm 0.011}$ | $0.164_{\pm 0.009}$ | $0.099_{\pm 0.011}$ | $0.119_{\pm 0.007}$ | $0.159_{\pm 0.006}$ | $0.073_{\pm 0.005}$ |
| **NVLM-D-72B** | $0.134_{\pm 0.010}$ | $0.179_{\pm 0.009}$ | $0.076_{\pm 0.008}$ | $0.085_{\pm 0.006}$ | $0.120_{\pm 0.007}$ | $0.053_{\pm 0.004}$ |
| **Qwen2.5-VL-72B-Instruct** | $0.207_{\pm 0.013}$ | $0.245_{\pm 0.011}$ | $0.133_{\pm 0.011}$ | $0.194_{\pm 0.007}$ | $0.227_{\pm 0.006}$ | $0.110_{\pm 0.006}$ |
| **QVQ-72B-Preview** | $0.197_{\pm 0.012}$ | $0.218_{\pm 0.010}$ | $0.091_{\pm 0.008}$ | $0.195_{\pm 0.007}$ | $0.215_{\pm 0.007}$ | $0.108_{\pm 0.006}$ |
| **llava-onevision-72b-ov-chat** | $0.141_{\pm 0.012}$ | $0.165_{\pm 0.010}$ | $0.097_{\pm 0.009}$ | $0.112_{\pm 0.008}$ | $0.141_{\pm 0.007}$ | $0.075_{\pm 0.005}$ |
| **gemma-3-27b-it** | $0.158_{\pm 0.011}$ | $0.218_{\pm 0.011}$ | $0.124_{\pm 0.010}$ | $0.106_{\pm 0.009}$ | $0.160_{\pm 0.009}$ | $0.075_{\pm 0.005}$ |
| **Aria** | $0.061_{\pm 0.009}$ | $0.101_{\pm 0.007}$ | $0.051_{\pm 0.006}$ | $0.035_{\pm 0.006}$ | $0.070_{\pm 0.006}$ | $0.046_{\pm 0.004}$ |
| **MiniCPM-V-2_6** | $0.043_{\pm 0.007}$ | $0.085_{\pm 0.006}$ | $0.064_{\pm 0.008}$ | $0.023_{\pm 0.004}$ | $0.057_{\pm 0.004}$ | $0.040_{\pm 0.003}$ |
| **Qwen2.5-VL-3B-Instruct** | $0.013_{\pm 0.003}$ | $0.040_{\pm 0.004}$ | $0.038_{\pm 0.006}$ | $0.014_{\pm 0.002}$ | $0.034_{\pm 0.003}$ | $0.023_{\pm 0.003}$ |
| *Proprietary LLMs* | | | | | | |
| **o3-mini-high** ♀ | $0.587_{\pm 0.023}$ | $0.684_{\pm 0.021}$ | $0.891_{\pm 0.018}$ | $0.445_{\pm 0.011}$ | $0.520_{\pm 0.011}$ | $0.512_{\pm 0.007}$ |
| **Claude-3-7-Sonnet** ♀ | $0.617_{\pm 0.019}$ | $0.712_{\pm 0.017}$ | $0.754_{\pm 0.021}$ | $0.492_{\pm 0.013}$ | $0.542_{\pm 0.012}$ | $0.431_{\pm 0.014}$ |
| **Claude-3-7-Sonnet** | $0.482_{\pm 0.015}$ | $0.574_{\pm 0.014}$ | $0.472_{\pm 0.019}$ | $0.446_{\pm 0.011}$ | $0.485_{\pm 0.011}$ | $0.321_{\pm 0.011}$ |
| **GPT-4o-2024-11-20** | $0.410_{\pm 0.018}$ | $0.472_{\pm 0.018}$ | $0.288_{\pm 0.019}$ | $0.338_{\pm 0.011}$ | $0.369_{\pm 0.012}$ | $0.196_{\pm 0.010}$ |
| **Gemini 2.0 Pro Exp** | $0.460_{\pm 0.014}$ | $0.525_{\pm 0.012}$ | $0.388_{\pm 0.016}$ | $0.425_{\pm 0.009}$ | $0.457_{\pm 0.008}$ | $0.289_{\pm 0.010}$ |
| **Gemini 2.0 Flash** | $0.301_{\pm 0.014}$ | $0.318_{\pm 0.012}$ | $0.255_{\pm 0.014}$ | $0.280_{\pm 0.007}$ | $0.298_{\pm 0.006}$ | $0.198_{\pm 0.006}$ |
| *Open-Weight LLMs* | | | | | | |
| **Llama-3.1-405B-Instruct**[†] | $0.161_{\pm 0.013}$ | $0.359_{\pm 0.012}$ | $0.243_{\pm 0.017}$ | $0.355_{\pm 0.013}$ | $0.390_{\pm 0.009}$ | $0.222_{\pm 0.008}$ |
| **DeepSeek-R1** ♀ | $0.646_{\pm 0.019}$ | $0.707_{\pm 0.017}$ | $0.678_{\pm 0.023}$ | $0.472_{\pm 0.011}$ | $0.507_{\pm 0.011}$ | $0.356_{\pm 0.011}$ |
| **DeepSeek-V3**[†] | $0.303_{\pm 0.014}$ | $0.369_{\pm 0.014}$ | $0.186_{\pm 0.013}$ | $0.290_{\pm 0.009}$ | $0.335_{\pm 0.008}$ | $0.145_{\pm 0.007}$ |
| **R1-Distill-Llama-70B** ♀ | $0.387_{\pm 0.015}$ | $0.448_{\pm 0.015}$ | $0.347_{\pm 0.017}$ | $0.285_{\pm 0.009}$ | $0.319_{\pm 0.009}$ | $0.161_{\pm 0.008}$ |
| **Llama-3.3-70B-Instruct** | $0.303_{\pm 0.013}$ | $0.371_{\pm 0.012}$ | $0.206_{\pm 0.014}$ | $0.280_{\pm 0.011}$ | $0.340_{\pm 0.009}$ | $0.173_{\pm 0.008}$ |
| **QwQ-32B** ♀ | $0.347_{\pm 0.017}$ | $0.445_{\pm 0.018}$ | $0.518_{\pm 0.020}$ | $0.254_{\pm 0.009}$ | $0.307_{\pm 0.009}$ | $0.189_{\pm 0.009}$ |
| **Open-Reasoner-Zero-32B** ♀ | $0.139_{\pm 0.010}$ | $0.204_{\pm 0.010}$ | $0.184_{\pm 0.012}$ | $0.146_{\pm 0.007}$ | $0.199_{\pm 0.007}$ | $0.095_{\pm 0.005}$ |
| **Phi-4** | $0.122_{\pm 0.010}$ | $0.194_{\pm 0.010}$ | $0.113_{\pm 0.011}$ | $0.140_{\pm 0.007}$ | $0.200_{\pm 0.006}$ | $0.085_{\pm 0.005}$ |

**Non-reasoning LLMs show limitations in ICR, and performance declines further with increasing grid size.** Among non-reasoning LLMs, Claude 3.7 Sonnet and Gemini 2.0 Pro Exp yield the best results, with WCRs of 0.482 and 0.460 on the 7x7 grid, respectively; however, their relatively lower ICR indicates model limitations on explicit reasoning constraints for crossword puzzles. Notably, thinking mode improves Claude 3.7 Sonnet on all three metrics, highlighting the importance of reasoning and reflection in solving constraints-based crossword tasks. Additionally, larger grids lead to decreasing performance, demonstrating the increased complexity of maintaining constraint adherence over a larger search space.

**LVLMs currently lag behind LLMs in performance, with minimal adherence to grid constraints.** With image inputs, Claude 3.7 Sonnet achieves the highest performance among LVLMs, but underperforms its own text inputs version. Models such as GPT-4o and Gemini 2.0 Pro Exp exhibit similar trends, with WCRs below 0.35 on larger grids. All LVLMs demonstrate low ICRs, suggesting that they struggle to maintain reasoning consistency. Notably, performance declines when thinking mode is enabled on Claude 3.7 Sonnet with image input, which contradicts the improvements observed with text inputs; we explore this phenomenon in the next section. Among open-weight LVLMs, Pixtral-Large-Instruct achieves the best WCR of 0.297 on 7x7 grid, while still lags behind most proprietary LVLMs.

### 4.3 Positive Correlation: LVLMs' Grid Parsing & Puzzle-Solving Performance

**Model performance on crossword puzzles exhibits a strong dependence on grid parsing capabilities, with systematic biases in word orientations.** We evaluate grid parsing ability by prompting models to parse completed puzzle grids and associated clues (prompt details are in Appendix D.4). Successful grid parsing requires: (1) identifying grid indexing numbers, (2) mapping numbers to clues to determine word orientation, and (3) extracting words with boundary recognition. The grid parsing ability reflects an LVLM's ability to interpret both spatial and textual information.

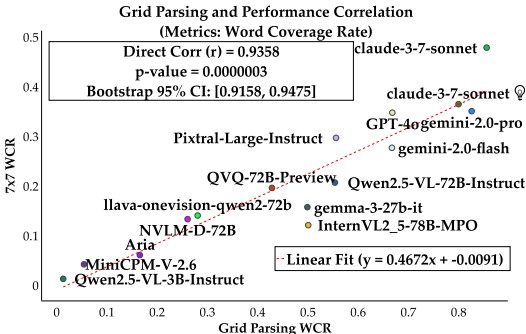

Figure 3: Grid Parsing vs. Puzzle-Solving on 7×7 English puzzles, measured with WCR.

We reveal two significant patterns: (1) As shown in Table 3, LVLMs extract Across words more accurately than Down words, highlighting OCR limitations. (2) Figure 3 demonstrates a strong positive correlation between grid parsing and overall puzzle-solving performance (both measured by WCR). Notably, when enabling thinking mode, Claude 3.7 Sonnet exhibits lower grid parsing performance, which aligns with its lower puzzle-solving performance. We hypothesize that the additional token generation in thinking mode diverts atten-

Table 3: WCR on Grid Parsing.

| Models | Across | Down |
|---|---|---|
| *Proprietary VLMs* | | |
| **Claude-3-7-Sonnet** | $0.954_{\pm 0.009}$ | $0.760_{\pm 0.018}$ |
| **Claude-3-7-Sonnet** ♀ | $0.949_{\pm 0.010}$ | $0.654_{\pm 0.022}$ |
| **GPT-4o-2024-11-20** | $0.886_{\pm 0.014}$ | $0.448_{\pm 0.024}$ |
| *Open-Weight VLMs* | | |
| **Pixtral-Large-Instruct** | $0.753_{\pm 0.022}$ | $0.361_{\pm 0.022}$ |
| **QVQ-72B-Preview** | $0.717_{\pm 0.022}$ | $0.139_{\pm 0.019}$ |

tion from image processing. These findings highlight the critical role of precise spatial-textual interpretation for future LVLM design. For full results, please refer to Appendix C.2.

### 4.4 Agentic Evaluation Setting: Interactive Mode for LVLMs

Motivated by recent research on visual-of-thoughts (Wu et al., 2024; Li et al., 2025), we introduce a new evaluation setting for LVLMs, referred to as **Interactive Mode**. This setting leverages the ability of CrossWordBench to automatically generate updated grid images.

**Interactive Mode** requires step-by-step puzzle solving rather than completing the entire puzzle in a single pass. Specifically, the implementation of the controllable generation framework allows for updating grid images each time a model provides an answer.

We maintain a four-round conversation history due to context window limitations. We introduce **Interactive Success Step (ISS)** to quantify how many words a model correctly solves before making its first error. Figure 4 shows the cumulative distribution of ISS for each model—**most models fail at the initial solution step on most puzzles**.

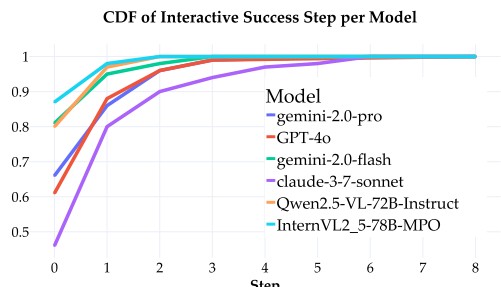

Figure 4: CDF of ISS on 7x7 English Puzzles.

Unlike the visual-of-thoughts approach, which focuses on generating intermediate reasoning plots, we employ external functions to update the grid. This establishes a foundation for evaluating LVLMs in agentic settings, where the word placement functions can serve as callable tools for the model and provide error feedback to guide the model in refining its responses—for example, by indicating whether a proposed word matches the expected length and whether its intersecting characters align with previously filled cells.

## 4.5 Beyond English: Evaluations on Multilingual, Dictionary-based, and Adapted Data

Table 4: WCR and ICR for Chinese, English Simple, and CommonSenseQA puzzle sets.

| Models | Chinese | | English Simple | | CommonSenseQA | |
|---|---|---|---|---|---|---|
| | WCR | ICR | WCR | ICR | WCR | ICR |
| *Proprietary LVLMs* | | | | | | |
| **GPT-4o-2024-11-20** | $0.366_{\pm0.018}$ | $0.170_{\pm0.017}$ | $0.335_{\pm0.015}$ | $0.227_{\pm0.017}$ | $0.392_{\pm0.026}$ | $0.247_{\pm0.026}$ |
| **Gemini 2.0 Flash** | $0.208_{\pm0.031}$ | $0.233_{\pm0.018}$ | $0.229_{\pm0.014}$ | $0.216_{\pm0.013}$ | $0.327_{\pm0.021}$ | $0.216_{\pm0.018}$ |
| *Open-Weight LVLMs* | | | | | | |
| **Pixtral-Large-Instruct-2411** | $0.252_{\pm0.015}$ | $0.101_{\pm0.011}$ | $0.216_{\pm0.016}$ | $0.187_{\pm0.015}$ | $0.439_{\pm0.023}$ | $0.270_{\pm0.023}$ |
| **Qwen2.5-VL-72B-Instruct** | $0.391_{\pm0.025}$ | $0.282_{\pm0.018}$ | $0.239_{\pm0.016}$ | $0.183_{\pm0.015}$ | $0.418_{\pm0.022}$ | $0.252_{\pm0.023}$ |
| *Proprietary LLMs* | | | | | | |
| **GPT-4o-2024-11-20** | $0.593_{\pm0.019}$ | $0.448_{\pm0.021}$ | $0.438_{\pm0.020}$ | $0.306_{\pm0.022}$ | $0.524_{\pm0.024}$ | $0.326_{\pm0.029}$ |
| **o3-mini-high** ♀ | $0.774_{\pm0.016}$ | $0.953_{\pm0.014}$ | $0.782_{\pm0.021}$ | $0.946_{\pm0.012}$ | $0.812_{\pm0.027}$ | $0.971_{\pm0.011}$ |
| *Open-Weight LLMs* | | | | | | |
| **DeepSeek-R1** ♀ | $0.907_{\pm0.016}$ | $0.898_{\pm0.018}$ | $0.759_{\pm0.017}$ | $0.787_{\pm0.020}$ | $0.752_{\pm0.031}$ | $0.829_{\pm0.026}$ |
| **QwQ-32B** ♀ | $0.701_{\pm0.020}$ | $0.654_{\pm0.022}$ | $0.647_{\pm0.021}$ | $0.734_{\pm0.020}$ | $0.699_{\pm0.026}$ | $0.766_{\pm0.027}$ |

We extend zero-shot CoT prompting evaluation to Chinese, dictionary-based, and benchmark-adapted word–clue pairs in CrossWordBench, with results shown in Table 4.

**Reasoning models outperform conventional ones across metrics, and the performance gap between open-weight and proprietary models is reduced.** o3-mini maintains a high ICR across all three datasets, consistent with the main evaluation results. Furthermore, the performance gap between open-weight and proprietary reasoning models is reduced, possibly due to the simplicity of these tasks and differences in training data. Interestingly, when solving Chinese puzzles, we observe that QwQ-32B consistently reasons in Chinese.

**Effectiveness of grid construction: puzzles constructed with CommonsenseQA training data remain challenging.** Despite CommonsenseQA having saturated performance on most

LLMs, its training set continues to pose significant challenges for reasoning models. For example, o3-mini achieves the highest WCR of 0.812, yet it still falls short of perfection. In contrast, the best-performing non-reasoning LLMs such as GPT-4o and open-weight LVLMs Pixtral-Large-Instruct obtain WCR of 0.524 and 0.439, respectively. These results highlight the effectiveness of our grid construction strategy in repurposing existing benchmark data.

## 5 Analysis

In this section, we provide an analysis of model behavior on CrossWordBench through the impact of structural constraints and reasoning mechanisms, and discuss future applications.

### 5.1 Grid Format Ablations: LLMs Robustness to Textual Grid Representations

**LLMs exhibit robustness to variations in grid text representation.** In the text-only evaluation setting, the empty crossword puzzle grid is represented as a 2D binary array, where 1 denotes a blocked cell and 0 denotes an unfilled cell. This array is then prepended to the prompt containing clues. To evaluate the impact of different formatting choices, we test an alternative markdown-style representation in this section, with · indicates unfilled cells and – indicates blocked cells. An example of the two formats is provided in Appendix D.5 and Figure 19.

Table 5: Grid Format Ablations.

| Models | Array | Markdown |
|---|---|---|
| **Claude-3-7-Sonnet** | $0.482_{\pm 0.015}$ | $0.760_{\pm 0.018}$ |
| **GPT-4o-2024-11-20** | $0.410_{\pm 0.014}$ | $0.398_{\pm 0.024}$ |
| **Gemini 2.0 Flash** | $0.301_{\pm 0.014}$ | $0.309_{\pm 0.015}$ |
| **DeepSeek-V3** | $0.303_{\pm 0.014}$ | $0.294_{\pm 0.016}$ |
| **o3-mini** | $0.587_{\pm 0.023}$ | $0.592_{\pm 0.024}$ |

Due to space constraints in the main text, we defer additional ablation studies on LVLM input formats and few-shot prompting to Appendix C.6 and C.8, respectively. We also demonstrate how controlling prefill ratio can be used to adjust task difficulty. We encourage readers to refer to these appendices for a comprehensive analysis and further discussion.

### 5.2 Crossing Letters: Reasoning LLMs Improve with More Intersections

**Average word accuracy on reasoning models increases with crossing letter count.**

Figure 5 presents the average WCR for each range of crossing letter counts, divided into three groups, across five reasoning and nine non-reasoning LLMs. We observe that reasoning models exhibit increasing accuracy with a greater number of letter intersections, whereas this trend is not observed for non-reasoning LLMs. This observation aligns with our main experiment results, in which reasoning models tend to exhibit higher ICRs, suggesting that they benefit from effectively utilizing grid constraints for solution space reduction. However, we do not observe the same pattern on 14x14 puzzles—likely because puzzles with larger grids are substantially more difficult even for reasoning models. We provide detailed results in Appendix C.7.

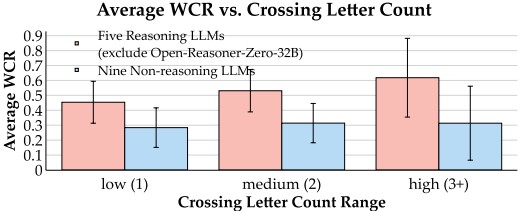

Figure 5: Crossing letter counts and average WCR on 7x7 English puzzles across LLMs.

To further isolate the effect of crossing-letter density within a constrained grid, we perform a quasi-ablation study on 7x7 puzzles by stratifying puzzles into low, medium, and high-density groups based on the average number of crossing letters per word, as shown in Table 6. Overall, within different densities, reasoning LLMs demonstrate improved WCR with a greater number of crossing letters, whereas non-reasoning LLMs show relatively stable performance.

Table 6: Average WCR results by crossing letter count and density on 7x7 English puzzles.

| Density | Crossing Letter | Reason. | Non-Reason. |
|---|---|---|---|
| Low | Low(1) | 0.4341 | 0.2960 |
| | Med(2) | 0.5275 | 0.3355 |
| | High(3+) | 0.6165 | 0.2794 |
| Medium | Low(1) | 0.4271 | 0.2754 |
| | Med(2) | 0.5477 | 0.3089 |
| | High(3+) | 0.6029 | 0.4193 |
| High | Low(1) | 0.5288 | 0.3097 |
| | Med(2) | 0.5151 | 0.3217 |
| | High(3+) | 0.5928 | 0.3030 |

### 5.3 Self-Reflection: Limited Impact on Crossword Puzzle Solving

Backtracking and verifying previously filled answers are essential components of effective crossword puzzle solving. To evaluate the effect of self-reflection, we include a manually crafted follow-up query that prompts models to revisit their previous zero-shot CoT responses (please see Appendix D.4 for details). As shown in Figure 6, we observe no measurable performance improvement for reasoning LLMs or non-reasoning LVLMs. This find suggests that additional interactions with manual prompting alone are insufficient to enhance reasoning capabilities for puzzle solving.

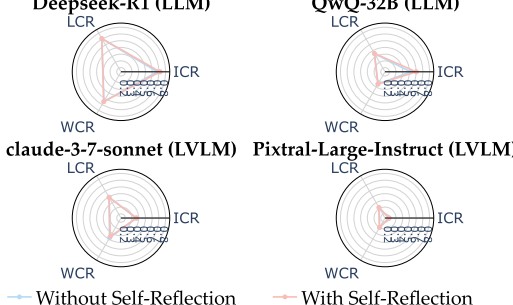

Figure 6: Self-reflection improvements on 7x7 English puzzles across metrics. Top: reasoning LLMs. Bottom: non-reasoning LVLMs.

### 5.4 Test-Time Scaling: Diminishing Returns on Puzzle Performance

To examine the effect of test-time scaling on CrossWordBench, we evaluate o3-mini on 7x7 English puzzles by varying its reasoning effort, which controls the number of reasoning tokens generated during inference. As shown in Figure 7, increasing the effort from low to medium yields a substantial performance improvement across all three metrics. However, further doubling the reasoning tokens provides no significant additional gains, indicating diminishing returns.

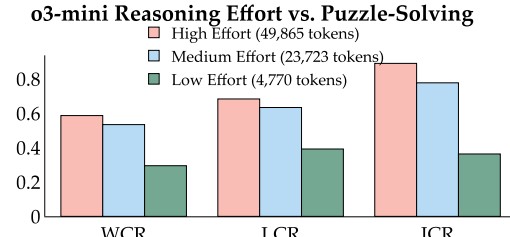

Figure 7: o3-mini performance on 7x7 English puzzles under three levels of reasoning efforts.

### 5.5 Discussion: Verifiable Crossword for Future Multimodal RL Training

Table 2 shows that reasoning models such as Open-Reasoner-Zero-32B, which are primarily trained on mathematical problems, exhibit limited generalization to CrossWordBench. This highlights the limitations of relying predominantly on math-based tasks with verifiable answers for reinforcement learning in more complex reasoning settings. As multimodal reasoning advances, a significant challenge persists: the lack of multimodal environments with verifiable answers suitable for rule-based reinforcement learning. We propose crossword puzzles as a compelling alternative, owing to their unique combinations of verifiability, multimodal structure, and goal-oriented task design. Crossword puzzles are also suitable for multi-turn training, where each word fill can represent a discrete, observable decision.

Our framework supports this setting by providing word placement functions that dynamically update the puzzle grid in both text and image formats, allowing seamless integration with multimodal agents and enabling tool-use training. We leave the exploration of training multimodal agents on crossword puzzles with real-time function feedback for future work.

## 6 Conclusion

This paper introduces CrossWordBench, a benchmark designed to evaluate the multimodal reasoning capabilities of both LLMs and LVLMs using crossword puzzles, which uniquely integrate text-based clues and visual constraints. Our extensive evaluation of over 20 models shows that reasoning models substantially outperform non-reasoning counterparts and can benefit from increased crossing-letter constraints. Additionally, we find a strong correlation between puzzle-solving performance and grid-parsing accuracy in LVLMs. Even puzzles derived from saturated benchmarks remain challenging, emphasizing the necessity of structural complexity in rigorous reasoning evaluation. This work paves the way for improving future multimodal RL training where interplay between modalities is essential.

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

# A  Acknowledgment

We thank all reviewers for their feedback, which contributed to improving this work.

# B  Limitations

We propose CrossWordBench to evaluate the reasoning capabilities of both LLMs and LVLMs. Unlike prior work, CrossWordBench does not incorporate any human-authored puzzles from online sources. Although our generation process is carefully designed to preserve key structural properties—such as clue distribution, word length, and blocked cell ratios—as summarized in Table 1, and the generated puzzles exhibit difficulty levels comparable to those of human—created puzzles when evaluated using standard word-level metrics, manual verification of puzzle quality could be helpful. Although our framework is also capable of generating crossword puzzles for potential use in training, this paper focuses solely on evaluation and leaves the exploration of reinforcement learning for future work.

# C  More Results and Analysis

## C.1  Behavior Analysis

To better analyze the errors made by models in puzzle-solving, we define two metrics: global length error and local length error. The global length error metric compares the number of words produced by the model with those in the reference answer list, assessing whether the model supplies an answer for every clue in the puzzle. In contrast, the local length error metric compares the length of each individual word to its corresponding reference, thereby quantifying the model's adherence to the grid constraints. Table 7 shows that even the best-performing reasoning models, such as Claude 3.7 Sonnet with thinking mode and DeepSeek R1, exhibit global length errors on two puzzles. Almost all models—with the exception of o3-mini, which demonstrates the highest ICR—commit a significant number of local errors. Moreover, we observe that all global length errors arise from models providing fewer answers than required, i.e., failing to address some clues. In contrast, most local length errors are due to models generating answers that exceed the expected word length.

## C.2  Complete Results on Grid Parsing Performance

In Section 4.3, we demonstrate a positive correlation between LVLM's grid-parsing accuracy and their puzzle-solving performance. We also provide several examples illustrating the limitations of LVLMs in extracting horizontal words. Here, we present a complete table of all models shown in Table 3, as summarized in Table 8, where a similar trend is observed.

## C.3  Complete Results on Extended Data

In this section, we present additional results for evaluations on the extended data. As shown in Table 9, LLMs generally perform better across these three categories than on the English puzzles; however, even for puzzles from CommonSenseQA, performance remains far from perfect. In contrast, LVLMs do not exhibit a significant performance improvement over English puzzles. Based on our observation of a strong positive correlation between grid-parsing and puzzle-solving performance (See Figure 3 for more details), we hypothesize that the reasoning capabilities of LVLMs are constrained by their visual processing abilities.

## C.4  More on Test-Time Scaling

In Section 5.4, we use a bar chart to demonstrate the performance differences associated with three distinct reasoning efforts for o3-mini on 7x7 English puzzles. Here, we provide a table detailing the specific numerical values for each metric as a supplement to Figure 7.

Table 7: Global and Local Length Errors across models on 7x7 English puzzles. "Long" and "Short" indicate words that are longer or shorter than the corresponding reference answer.

| Models | Global Length Error | | | Local Length Error | | |
|---|---|---|---|---|---|---|
| | Tot. | Long | Short | Tot. | Long | Short |
| *Proprietary LVLMs* | | | | | | |
| Claude-3-7-Sonnet | 0 | 0 | 0 | 363 | 205 | 158 |
| Claude-3-7-Sonnet ♀ | 0 | 0 | 0 | 454 | 237 | 217 |
| GPT-4o-2024-11-20 | 10 | 0 | 10 | 581 | 326 | 255 |
| Gemini 2.0 Pro Exp | 1 | 0 | 1 | 565 | 467 | 98 |
| Gemini 2.0 Flash | 0 | 0 | 0 | 665 | 568 | 97 |
| *Open-Weight LVLMs* | | | | | | |
| Pixtral-Large-Instruct-2411 | 3 | 0 | 3 | 623 | 481 | 142 |
| InternVL2_5-78B-MPO | 0 | 0 | 0 | 834 | 682 | 152 |
| NVLM-D-72B | 8 | 0 | 8 | 791 | 620 | 171 |
| Qwen2.5-VL-72B-Instruct | 2 | 0 | 2 | 744 | 600 | 144 |
| QVQ-72B-Preview | 20 | 0 | 20 | 765 | 525 | 240 |
| llava-onevision-72b-ov-chat | 3 | 0 | 3 | 829 | 715 | 114 |
| gemma-3-27b-it | 18 | 0 | 18 | 781 | 645 | 136 |
| Aria | 16 | 0 | 16 | 894 | 720 | 174 |
| MiniCPM-V-2_6 | 16 | 0 | 16 | 918 | 688 | 230 |
| Qwen2.5-VL-3B-Instruct | 31 | 0 | 31 | 1034 | 613 | 421 |
| *Proprietary LLMs* | | | | | | |
| o3-mini ♀ | 0 | 0 | 0 | 6 | 4 | 2 |
| Claude-3-7-Sonnet ♀ | 2 | 0 | 2 | 124 | 44 | 80 |
| Claude-3-7-Sonnet | 1 | 0 | 1 | 274 | 74 | 200 |
| GPT-4o-2024-11-20 | 17 | 0 | 17 | 399 | 183 | 216 |
| Gemini 2.0 Pro Exp | 0 | 0 | 0 | 378 | 254 | 124 |
| Gemini 2.0 Flash | 0 | 0 | 0 | 633 | 545 | 88 |
| *Open-Weight LLMs* | | | | | | |
| Llama-3.1-405B-Instruct[†] | 8 | 0 | 8 | 835 | 741 | 94 |
| DeepSeek-R1 ♀ | 2 | 0 | 2 | 25 | 14 | 11 |
| DeepSeek-V3[†] | 11 | 0 | 11 | 513 | 206 | 307 |
| R1-Distill-Llama-70B ♀ | 9 | 0 | 9 | 203 | 110 | 93 |
| Llama-3.3-70B-Instruct | 22 | 0 | 22 | 598 | 326 | 272 |
| QwQ-32B ♀ | 9 | 0 | 9 | 65 | 34 | 31 |
| Open-Reasoner-Zero-32B ♀ | 10 | 0 | 10 | 697 | 473 | 224 |
| Phi-4 | 2 | 0 | 2 | 709 | 447 | 262 |

Table 8: WCR of Grid Parsing for all models.

| Models | Size | Across | Down | All |
|---|---|---|---|---|
| *Proprietary LVLMs* | | | | |
| Claude-3-7-Sonnet | - | $0.954_{\pm0.009}$ | $0.760_{\pm0.018}$ | $0.855_{\pm0.010}$ |
| Claude-3-7-Sonnet ♀ | - | $0.949_{\pm0.010}$ | $0.654_{\pm0.022}$ | $0.800_{\pm0.012}$ |
| GPT-4o-2024-11-20 | - | $0.886_{\pm0.014}$ | $0.448_{\pm0.024}$ | $0.668_{\pm0.015}$ |
| Gemini 2.0 Pro Exp | - | $0.962_{\pm0.008}$ | $0.693_{\pm0.018}$ | $0.826_{\pm0.011}$ |
| Gemini 2.0 Flash | - | $0.954_{\pm0.009}$ | $0.381_{\pm0.024}$ | $0.667_{\pm0.013}$ |
| *Open-Weight LVLMs* | | | | |
| Pixtral-Large-Instruct-2411 | 124B | $0.753_{\pm0.022}$ | $0.361_{\pm0.022}$ | $0.556_{\pm0.016}$ |
| NVLM-D-72B | 78.4B | $0.429_{\pm0.024}$ | $0.099_{\pm0.015}$ | $0.261_{\pm0.013}$ |
| InternVL2_5-78B-MPO | 78.4B | $0.744_{\pm0.019}$ | $0.258_{\pm0.021}$ | $0.501_{\pm0.014}$ |
| Qwen2.5-VL-72B-Instruct | 73.4B | $0.730_{\pm0.017}$ | $0.378_{\pm0.023}$ | $0.554_{\pm0.015}$ |
| QVQ-72B-Preview | 73.4B | $0.717_{\pm0.022}$ | $0.139_{\pm0.019}$ | $0.428_{\pm0.017}$ |
| llava-onevision-qwen2-72b-ov-chat | 73.2B | $0.382_{\pm0.021}$ | $0.185_{\pm0.020}$ | $0.281_{\pm0.014}$ |
| gemma-3-27b-it | 27.4B | $0.746_{\pm0.021}$ | $0.250_{\pm0.021}$ | $0.499_{\pm0.015}$ |
| Aria | 25.3B | $0.258_{\pm0.022}$ | $0.074_{\pm0.014}$ | $0.165_{\pm0.013}$ |
| MiniCPM-V-2_6 | 8.1B | $0.091_{\pm0.015}$ | $0.018_{\pm0.006}$ | $0.054_{\pm0.009}$ |
| Qwen2.5-VL-3B-Instruct | 3.75B | $0.023_{\pm0.007}$ | $0.003_{\pm0.002}$ | $0.013_{\pm0.004}$ |

Table 9: WCR and ICR for Chinese, English Simple, and CommonSenseQA puzzle sets.

| Models | Chinese | | English Simple | | CommonSenseQA | |
|---|---|---|---|---|---|---|
| | WCR | ICR | WCR | ICR | WCR | ICR |
| *Proprietary LVLMs* | | | | | | |
| GPT-4o-2024-11-20 | $0.366_{\pm0.018}$ | $0.170_{\pm0.017}$ | $0.335_{\pm0.015}$ | $0.227_{\pm0.017}$ | $0.392_{\pm0.026}$ | $0.247_{\pm0.026}$ |
| Claude 3.7 sonnet | $0.339_{\pm0.023}$ | $0.267_{\pm0.018}$ | $0.408_{\pm0.018}$ | $0.288_{\pm0.018}$ | $0.540_{\pm0.022}$ | $0.386_{\pm0.028}$ |
| Gemini 2.0 Flash | $0.208_{\pm0.031}$ | $0.233_{\pm0.018}$ | $0.229_{\pm0.014}$ | $0.216_{\pm0.013}$ | $0.327_{\pm0.021}$ | $0.216_{\pm0.018}$ |
| *Open-Weight LVLMs* | | | | | | |
| Pixtral-Large-Instruct-2411 | $0.252_{\pm0.015}$ | $0.101_{\pm0.011}$ | $0.216_{\pm0.016}$ | $0.187_{\pm0.015}$ | $0.439_{\pm0.023}$ | $0.270_{\pm0.023}$ |
| Qwen2.5-VL-72B-Instruct | $0.391_{\pm0.025}$ | $0.282_{\pm0.018}$ | $0.239_{\pm0.016}$ | $0.183_{\pm0.015}$ | $0.418_{\pm0.022}$ | $0.252_{\pm0.023}$ |
| *Proprietary LLMs* | | | | | | |
| GPT-4o-2024-11-20 | $0.593_{\pm0.019}$ | $0.448_{\pm0.021}$ | $0.438_{\pm0.020}$ | $0.306_{\pm0.022}$ | $0.524_{\pm0.024}$ | $0.326_{\pm0.029}$ |
| Claude 3.7 sonnet | $0.478_{\pm0.019}$ | $0.470_{\pm0.019}$ | $0.539_{\pm0.021}$ | $0.487_{\pm0.021}$ | $0.583_{\pm0.024}$ | $0.518_{\pm0.025}$ |
| o3-mini-high ♀ | $0.774_{\pm0.016}$ | $0.953_{\pm0.014}$ | $0.782_{\pm0.021}$ | $0.946_{\pm0.012}$ | $0.812_{\pm0.027}$ | $0.971_{\pm0.011}$ |
| *Open-Weight LLMs* | | | | | | |
| DeepSeek-R1 ♀ | $0.907_{\pm0.016}$ | $0.898_{\pm0.018}$ | $0.759_{\pm0.017}$ | $0.787_{\pm0.020}$ | $0.752_{\pm0.031}$ | $0.829_{\pm0.026}$ |
| QwQ-32B ♀ | $0.701_{\pm0.020}$ | $0.654_{\pm0.022}$ | $0.647_{\pm0.021}$ | $0.734_{\pm0.020}$ | $0.699_{\pm0.026}$ | $0.766_{\pm0.027}$ |

Table 10: o3-mini performance on 7x7 English puzzles across three distinct reasoning efforts.

| Reasoning Effort | WCR | LCR | ICR | Avg. Tokens |
|---|---|---|---|---|
| High | $0.587_{\pm0.023}$ | $0.684_{\pm0.021}$ | $0.891_{\pm0.018}$ | 49865 |
| Medium | $0.534_{\pm0.022}$ | $0.634_{\pm0.022}$ | $0.777_{\pm0.025}$ | 23723 |
| Low | $0.295_{\pm0.018}$ | $0.392_{\pm0.019}$ | $0.363_{\pm0.026}$ | 4770 |

## C.5 Token Usage

In this section, we report the token usage for all evaluated models on both the 7x7 and 14x14 English puzzles. Notably, we include non-reasoning models in this analysis, defining token usage as the total number of completion tokens. For reasoning models, token usage is calculated as the sum of reasoning tokens and response tokens. As shown in Figure 8, token usage increases with grid size across all models, with reasoning models generating more.

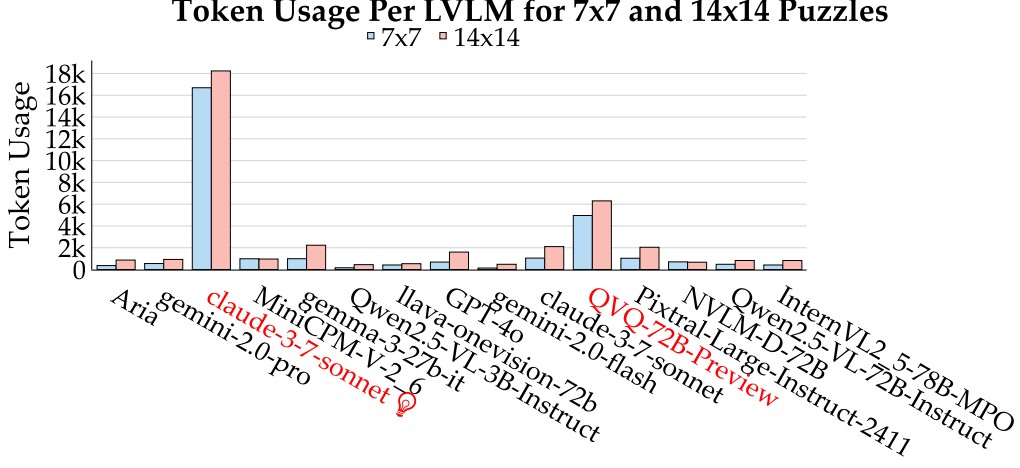

(a) Token Usage of LVLMs.

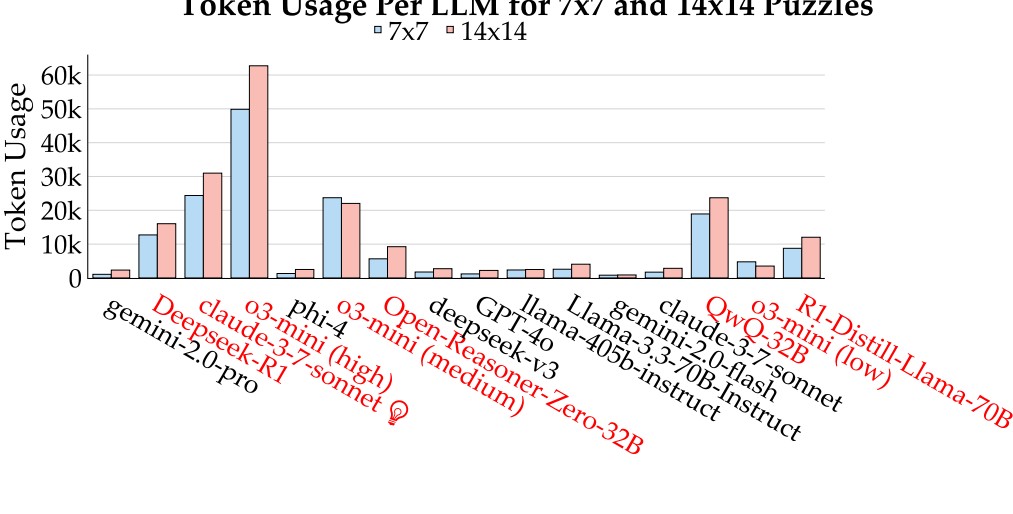

(b) Token Usage of LLMs.

Figure 8: Token usage on 7x7 and 14x14 English puzzles. Reasoning models are in red.

## C.6 Ablations: LVLM Input Format and Modality Effects

To isolate modality-specific effects, we explore feeding LLMs the OCR-extracted output from grid images. Our experiments show that while both OCR models and LVLMs can accurately extract clue text from images, they struggle to follow instructions and to construct a spatially coherent representation of the empty grid. For example, when applying stepfun-ai/GOT-OCR-2.0-hf to an image containing both the grid and clues, it generates *"Across:\n2. Canon*

*product, for short\n4. Defense aid\n6. Certain fraud protector, for short\n7. Abbr. before a founding date\n10. Philosopher's study\n12. May honoree\n13. Fraternity letter\nDown:\n1. Like white panthers\n2. [Not my mistake]\n3. Beta dog's view\n5. Gridiron abbr.\n8. One of the muskrats in the 1976 hit "Muskrat Love"\n9. Slow-witted\n11. Going rate?: Abbr.\n"*. While the clue text is correctly extracted, the output lacks any representation of the grid structure, rendering it unsuitable as input for LLMs due to the absence of critical structural constraints. On the other hand, the clue content extracted by OCR models and LVLMs is nearly identical to what we explicitly feed to LLMs in the text input setting. In fact, we augment the clue text with explicit positional hints to compensate for the lack of visual spatial information—details that LVLMs might infer from the image but LLMs cannot derive from plain text alone.

In our main evaluations, both the clues and the grid are embedded within a single image as inputs for LVLMs. In this section, we investigate the effect of isolating the clues and providing them to the LVLMs as separate text inputs, while the image contains only an empty grid. The prompt used for this experimental setting is shown in Figure 11 in Appendix D.

The results, shown in Figure 9 and measured by WCR on 7x7 English puzzles, reveal no significant performance differences between the two input formats. This indicates that the input format is not the primary cause of the suboptimal performance observed for LVLMs on CrossWordBench, suggesting that LVLMs are robust to slight variations in input format.

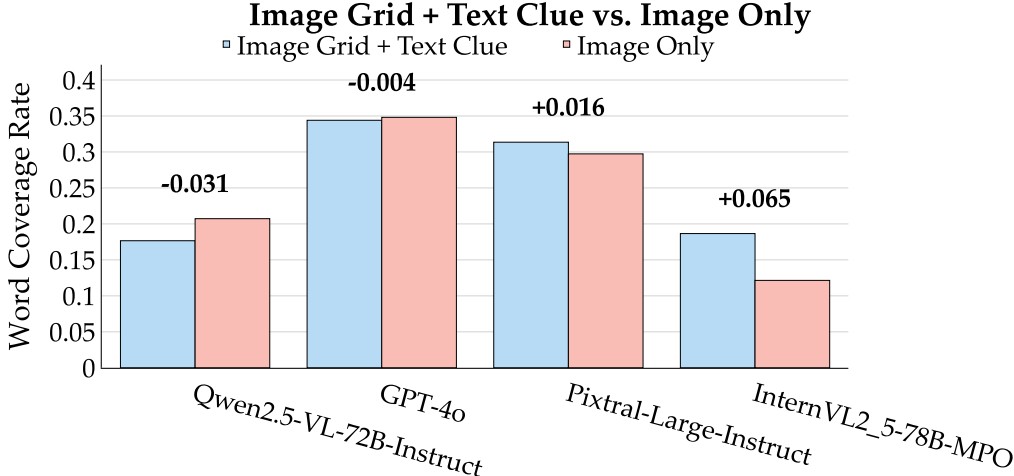

Figure 9: WCR difference on two inputs formats for LVLMs.

## C.7 More on Crossing Letter Count

Table 11 presents the results shown in Figure 5, along with additional results on 14x14 English puzzles. On these larger puzzles, even reasoning LLMs struggle to effectively leverage crossing-letter constraints. We hypothesize that this is due to the increased complexity of larger puzzles, which remains challenging for reasoning models.

Table 11: Average WCR results by crossing letter count on 7x7 and 14x14 English puzzles.

| Grid | Crossing Letter | Reason. | Non-Reason. |
|------|-----------------|---------|-------------|
| 7x7 | Low | 0.4539 | 0.2922 |
| 7x7 | Med | 0.5308 | 0.3230 |
| 7x7 | High | 0.6183 | 0.3259 |
| 14x14 | Low | 0.4115 | 0.3469 |
| 14x14 | Med | 0.4186 | 0.3514 |
| 14x14 | High | 0.2306 | 0.1438 |

## C.8 More on Few-Shot Prompting and Prefill Ratio Control

In our main evaluation, we treat each puzzle as a whole rather than as a set of independent question–answer (QA) pairs, in order to preserve the structural constraints inherent to crossword solving. This setup is not well-suited to few-shot prompting, as providing a full puzzle solution as a demonstration example would be lengthy and potentially distracting for the model. Nevertheless, we experiment with one-shot prompting on 7x7 English puzzles,

using a completed example generated by DeepSeek-R1 on the English Simple subset as the demonstration. As shown in Figure 12a, the performance difference is not significant.

As discussed in Section 5.1, our controllable generation framework supports difficulty adjustment via the prefill ratio. In Table 12b, we compare performance on 7x7 English puzzles under two prefill conditions: 0% and 50%. To ensure a fair comparison, we prevent any word from being fully revealed. As expected, a higher prefill ratio leads to improved performance, as more letters are made available to the model. These results suggest that prefill ratio is a more effective method for puzzle difficulty control than few-shot prompting.

Table 12: WCR under different prompting and prefill ratio settings on 7x7 English puzzles.

(a) WCR with 0-shot and 1-shot prompting.

| Model | 0-Shot | 1-Shot |
|---|---|---|
| GPT-4o (Text) | 0.410 | 0.394 |
| Claude-Sonnet (Text) | 0.482 | 0.519 |
| Gemini-Flash (Text) | 0.301 | 0.313 |
| Llama-3.3-70B (Text) | 0.303 | 0.269 |
| DeepSeek-V3 (Text) | 0.303 | 0.307 |

(b) WCR with 0% and 50% prefill.

| Model | 0% | 50% |
|---|---|---|
| GPT-4o (Text) | 0.410 | 0.483 |
| GPT-4o (Img) | 0.348 | 0.383 |
| Claude-Sonnet (Text) | 0.482 | 0.604 |
| Claude-Sonnet (Img) | 0.479 | 0.568 |
| Gemini-Flash (Text) | 0.301 | 0.362 |
| Gemini-Flash (Img) | 0.277 | 0.346 |
| o3-mini (Text) | 0.587 | 0.864 |

# D   Implementation Details

## D.1   Evaluation Metrics

Here, we provide a more complete and formal description of the three metrics used.

- **Word Coverage Rate (WCR):** WCR measures word-level accuracy by calculating the percentage of correctly solved words in the crossword puzzle, defined as:

$$\text{WCR} = \frac{1}{|\mathcal{W}_A| + |\mathcal{W}_D|} \left( \sum_{w \in \mathcal{W}_A} \mathbb{1}\{r_w = m_w\} + \sum_{w \in \mathcal{W}_D} \mathbb{1}\{r_w = m_w\} \right). \tag{2}$$

  where $\mathcal{W}_A$ and $\mathcal{W}_D$ denote the set of across and down words, respectively. For each word $w$, $r_w$ represents the reference answer, while $m_w$ represents the model answer.

- **Letter Coverage Rate (LCR):** LCR evaluates letter-level accuracy, providing partial credit for correct letter placements. For each word $w$, Let:

$$C_w = \sum_{j=1}^{\min(|r_w|,|m_w|)} \mathbb{1}\{r_w[j] = m_w[j]\} \text{ and } L_w = \max(|r_w|, |m_w|)$$

  where $C_w$ counts the correctly matched letters and $L_w$ is the total number of positions considered, the overall letter accuracy is defined as:

$$\text{LCR} = \frac{\sum_{w \in \mathcal{W}} C_w}{\sum_{w \in \mathcal{W}} L_w} \tag{3}$$

- **Intersection Consistency Rate (ICR):** the internal consistency of the model's answers at intersections where across and down words overlap, defined as:

$$\text{ICR} = \frac{1}{|\mathcal{I}|} \sum_{(a,d,j,k) \in \mathcal{I}} \mathbb{1}\{a[j] = d[k]\} \tag{4}$$

  where $\mathcal{I}$ denotes the set of all intersections, where each tuple $(a, d, j, k)$ indicates the $j$th letter of the across word $a$ overlaps with the $k$th letter of the down word $d$. This metric reflects whether models correctly adhere to the grid structural constraints of a puzzle.

### D.2 Parsing Details

Answers extracted from model responses are converted into JSON format by leveraging the structured output capabilities of o3-mini and dynamic Pydantic models that adhere to the reference answer structure. Algorithm 1 provides the pseudocode for creating these models.

---

**Algorithm 1** Create Dynamic Pydantic Model

---

1: **function** CREATE_DYNAMIC_PYDANTIC_MODEL(reference_answers)
2:     fields ← {}                                      ▷ Initialize an empty dictionary
3:     pattern ← ^(across|down)\s+\d+\$
4:     **for all** $answer$ ∈ reference_answers **do**
5:         key ← $answer$["direction"]
6:         **if** key does not match pattern **then**
7:             **raise** ValueError("Reference key does not match expected format")
8:         **end if**
9:         fields[key] ← (Optional[ClueAnswer], Field with description "Answer for clue at key")
10:     **end for**
11:     Dynamic_Pydantic_Model ← create_model("Dynamic_Pydantic_Model", **fields)
12:     **return** Dynamic_Pydantic_Model
13: **end function**

---

### D.3 Generation Configuration

Table 13 lists all the models evaluated along with their corresponding generation configs.

| Model Name | Max Tokens (7×7) | Max Tokens (14×14) | Temperature |
|---|---|---|---|
| Claude 3.7 Sonnet (Anthropic, 2024) | 8192 | 8192 | 0.0 |
| Claude 3.7 Sonnet (Thinking) (Anthropic, 2024) | 64000 | 64000 | 1.0 |
| GPT-4o-2024-11-20 (Hurst et al., 2024) | 16384 | 16384 | 0.0 |
| Gemini 2.0 Pro Exp (DeepMind, 2024) | 20480 | 20480 | 0.0 |
| Gemini 2.0 Flash (DeepMind, 2024) | 20480 | 20480 | 0.0 |
| o3-mini-high (OpenAI, 2025) | 100000 | 100000 | 0.6 |
| Pixtral-Large-Instruct-2411 (Agrawal et al., 2024) | 20480 | 20480 | 0.0 |
| NVLM-D-72B (Dai et al., 2024) | 20480 | 20480 | 0.0 |
| InternVL2 5-78B-MPO (Chen et al., 2024) | 20480 | 20480 | 0.0 |
| Qwen2.5-VL-72B-Instruct (Bai et al., 2025) | 20480 | 20480 | 0.0 |
| QVQ-72B-Preview (Team, 2024) | 100000 | 100000 | 0.0 |
| llava-onevision-72b-ov-chat (Li et al., 2024a) | 20480 | 20480 | 0.0 |
| gemma-3-27b-it (Team et al., 2025) | 8192 | 8192 | 0.0 |
| Aria (Li et al., 2024b) | 20480 | 20480 | 0.0 |
| MiniCPM-V-2_6 (Yao et al., 2024) | 20480 | 20480 | 0.0 |
| Qwen2.5-VL-3B-Instruct (Bai et al., 2025) | 20480 | 20480 | 0.0 |
| Llama-3.1-405B-Instruct (Dubey et al., 2024) | 100000 | 100000 | 0.0 |
| DeepSeek-R1 (Guo et al., 2025) | 100000 | 100000 | 0.6 |
| DeepSeek-V3 (Liu et al., 2024) | 100000 | 100000 | 0.0 |
| R1-Distill-Llama-70B (Guo et al., 2025) | 100000 | 100000 | 0.6 |
| Llama-3.3-70B-Instruct (Dubey et al., 2024) | 20480 | 20480 | 0.0 |
| QwQ-32B (Team, 2025) | 100000 | 100000 | 0.6 |
| Phi-4 (Abdin et al., 2024) | 10000 | 10000 | 0.0 |

Table 13: Generation Configurations for CrossWordBench.

### D.4 Prompts

Figures 10, 11, 12, 13, 14, 15, 17, and 18 present all the prompts employed in this study—comprising image prompts for LVLMs and text prompts for LLMs. We observe that Claude 3.7 Sonnet sometimes produces partial outputs and requests confirmation to continue. To mitigate this issue, we incorporate an additional system prompt (see Figure 18 for the system prompt); note that this modification applies only to the Claude 3.7 Sonnet.

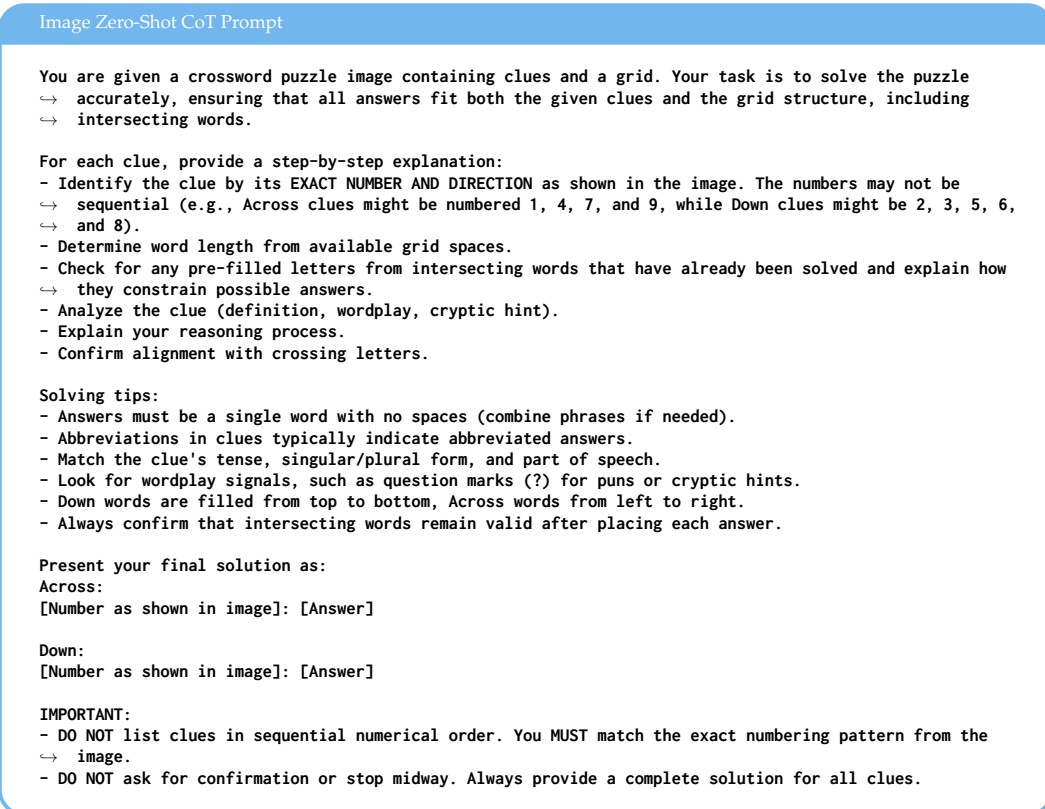

Figure 10: Image Zero-Shot CoT Prompt.

**Image Zero-Shot CoT with Grid Only Prompt**

```
You are given a crossword puzzle image containing only the grid, while the clues are provided separately
↪ in text form. Your task is to solve the puzzle accurately, ensuring that all answers fit both the
↪ given clues and the grid structure, including intersecting words.

Clues:
Each clue contains:
- Clue Direction and Number (e.g., "Across 1", "Down 2").
- Start Position (row, column) for the first letter of the answer.
- The actual clue text.

<clues>

For each clue, provide a step-by-step explanation:
- Identify the clue by its EXACT NUMBER AND DIRECTION as shown in the clue description. The numbers may
↪ not be sequential (e.g., Across clues might be numbered 1, 4, 7, and 9, while Down clues might be 2,
↪ 3, 5, 6, and 8).
- Determine word length from available grid spaces.
- Check for any pre-filled letters from intersecting words that have already been solved and explain how
↪ they constrain possible answers.
- Analyze the clue (definition, wordplay, cryptic hint).
- Explain your reasoning process.
- Confirm alignment with crossing letters.

Solving tips:
- Answers must be a single word with no spaces (combine phrases if needed).
- Abbreviations in clues typically indicate abbreviated answers.
- Match the clue's tense, singular/plural form, and part of speech.
- Look for wordplay signals, such as question marks (?) for puns or cryptic hints.
- Down words are filled from top to bottom, Across words from left to right.
- Always confirm that intersecting words remain valid after placing each answer.

Present your final solution as:
Across:
[Number as shown in clues]: [Answer]

Down:
[Number as shown in clues]: [Answer]

IMPORTANT:
- DO NOT list clues in sequential numerical order. You MUST match the exact numbering pattern from the
↪ given clues.
- DO NOT ask for confirmation or stop midway. Always provide a complete solution for all clues.
```

Figure 11: Image Zero-Shot CoT with Grid Only Prompt.

---

**Text Zero-Shot CoT Prompt**

```
You are given a crossword puzzle grid and a set of clues. Your task is to solve the puzzle accurately,
↪  ensuring that all answers fit both the given clues and the grid structure, including intersecting
↪  words.

Grid Representation:
The crossword grid is represented as a 2D array where:
- `1` represents a black (blocked) cell
- `0` represents an empty (unfilled) cell

<grid>

Clues:
Each clue contains:
- Clue Direction and Number (e.g., "Across 1", "Down 2").
- Start Position (row, column) for the first letter of the answer.
- The actual clue text.

<clues>

For each clue, provide a step-by-step explanation:
- Identify the clue by its EXACT NUMBER AND DIRECTION as shown in the clue description. The numbers may
↪  not be sequential (e.g., Across clues might be numbered 1, 4, 7, and 9, while Down clues might be 2,
↪  3, 5, 6, and 8).
- Determine word length from available grid spaces.
- Check for any pre-filled letters from intersecting words that have already been solved and explain how
↪  they constrain possible answers.
- Analyze the clue (definition, wordplay, cryptic hint).
- Explain your reasoning process.
- Confirm alignment with crossing letters.

Solving tips:
- Answers must be a single word with no spaces (combine phrases if needed).
- Abbreviations in clues typically indicate abbreviated answers.
- Match the clue's tense, singular/plural form, and part of speech.
- Look for wordplay signals, such as question marks (?) for puns or cryptic hints.
- Down words are filled from top to bottom, Across words from left to right.
- Always confirm that intersecting words remain valid after placing each answer.

Present your final solution as:
Across:
[Number as shown in clues]: [Answer]

Down:
[Number as shown in clues]: [Answer]

IMPORTANT:
- DO NOT list clues in sequential numerical order. You MUST match the exact numbering pattern from the
↪  given clues.
- DO NOT ask for confirmation or stop midway. Always provide a complete solution for all clues.
```

Figure 12: Text Zero-Shot CoT Prompt.

---

**Interactive Mode Prompt**

```
You are given a crossword puzzle image containing clues and a grid. Your task is to solve the puzzle
↪   accurately, ensuring that all answers fit both the given clues and the grid structure, including
↪   intersecting words.

Pick ONE clue, provide a step-by-step explanation:
- Identify the clue by its EXACT NUMBER AND DIRECTION as shown in the image. The numbers may not be
↪   sequential (e.g., Across clues might be numbered 1, 4, 7, and 9, while Down clues might be 2, 3, 5, 6,
↪   and 8).
- Determine word length from available grid spaces.
- Check for any pre-filled letters from intersecting words that have already been solved and explain how
↪   they constrain possible answers.
- Analyze the clue (definition, wordplay, cryptic hint).
- Explain your reasoning process.
- Confirm alignment with crossing letters.

Solving tips:
- Answers must be a single word with no spaces (combine phrases if needed).
- Abbreviations in clues typically indicate abbreviated answers.
- Match the clue's tense, singular/plural form, and part of speech.
- Look for wordplay signals, such as question marks (?) for puns or cryptic hints.
- Down words are filled from top to bottom, Across words from left to right.
- Always confirm that intersecting words remain valid after placing each answer.

Only solve ONE clue at a time and wait for confirmation before proceeding to the next round.
```

Figure 13: Interactive Mode Prompt.

---

**Interactive Mode Follow-up Prompt**

```
For the following round:

Using the confirmed answers so far:

Pick another clue, provide a step-by-step explanation:
- Identify the clue by its EXACT NUMBER AND DIRECTION as shown in the image. The numbers may not be
↪   sequential (e.g., Across clues might be numbered 1, 4, 7, and 9, while Down clues might be 2, 3, 5, 6,
↪   and 8).
- Determine word length from available grid spaces.
- Check for any pre-filled letters from intersecting words that have already been solved and explain how
↪   they constrain possible answers.
- Analyze the clue (definition, wordplay, cryptic hint).
- Explain your reasoning process.
- Confirm alignment with crossing letters.

Solving tips:
- Prioritize clues that intersect with confirmed answers.
- Answers must be a single word with no spaces (combine phrases if needed).
- Abbreviations in clues typically indicate abbreviated answers.
- Match the clue's tense, singular/plural form, and part of speech.
- Look for wordplay signals, such as question marks (?) for puns or cryptic hints.
- Down words are filled from top to bottom, Across words from left to right.
- Always confirm that intersecting words remain valid after placing each answer.

Only solve ONE clue at a time and wait for confirmation before proceeding to the next round and do not
↪   repeat previously solved clues.
```

Figure 14: Interactive Mode Follow-up Prompt.

---

**Grid-Parsing Prompt**

```
Your task is to extract and match all words from a crossword puzzle grid with their respective clues. The
↪  image consists of two sections:
1. The Clue Section:
- Contains two lists: "Across" and "Down."
- Each clue is numbered and corresponds to a starting position in the grid.
2. The Grid Section:
- A crossword grid containing letters, empty cells, and numbered starting positions for words.
- Words extend either across (left to right) or down (top to bottom).

Step 1: Extract Clues and Grid Structure
- Identify all clues under the "Across" and "Down" sections, preserving their numbers.
- Identify all numbered cells in the grid.

Step 2: Extract Words from the Grid
For each numbered cell:
- If the word extends ACROSS:
    - Start at the numbered cell and read consecutive letters left to right until reaching an empty cell
      ↪  or grid boundary.
- If the word extends DOWN:
    - Start at the numbered cell and read consecutive letters top to bottom until reaching an empty cell
      ↪  or grid boundary.

Step 3: Match Words to Clues
- Match each numbered word in the grid to its corresponding clue in the Across or Down section.
- Ensure extracted words are correctly assigned to their respective clues.

Output Format:
ACROSS:
[Number as shown in image]: [Clue Text]
Extracted Word: [Word from Grid]

DOWN:
[Number as shown in image]: [Clue Text]
Extracted Word: [Word from Grid]

Ensure accuracy in matching words to their clues, and extract all words fully without omitting any.
```

Figure 15: Grid-Parsing Prompt.

---

**Self-Reflection Prompt**

```
Your previous solution contains incorrect answers. Take a step back, carefully re-examine your entries,
↪  and systematically verify each word to ensure complete consistency and correctness within the
↪  crossword puzzle.

Provide a step-by-step verification:
1. Cross-Check Letters: List every intersection explicitly, noting the letters where Across and Down
↪  clues meet.
2. Consistency Check: Verify that each intersection matches perfectly. Identify and highlight any
↪  conflicting letters immediately.
3. Clue Validation: Revisit each clue thoroughly, confirming that each answer fully aligns with its clue
↪  description and adheres strictly to length constraints.
4. Grid Integrity: Confirm that your corrected entries maintain the integrity of the puzzle grid, leaving
↪  no unresolved conflicts or empty cells.

After completing these steps, present your revised and verified solutions in the following format:
Across:
[Clue Number]: [Corrected Answer]

Down:
[Clue Number]: [Corrected Answer]

IMPORTANT:
- DO NOT list clues in sequential numerical order. You MUST match the exact numbering pattern.
- Do NOT restate previous incorrect answers. Provide only fully corrected solutions after reflection.
```

Figure 16: Self-Reflection Prompt.

---

**Answer-Parsing System Prompt**

```
You are a crossword puzzle answer extractor. Extract only valid answers from a text response containing
↪  crossword solutions.

Requirements:
- If an answer contains spaces or multiple words, combine them into a single word.
- Do not shift or reorder answers. For example, if the expected keys are "Down 12" and "Down 13" and only
↪  the answer for "Down 13" is provided in the text, then "Down 12" should be null and "Down 13" should
↪  contain the answer.
- Do not invent or infer answers not explicitly stated in the text

Output your response in the given structure.
```

Figure 17: Answer-Parsing Prompt.

---

**Claude System Prompt**

```
You are a helpful assistant who completes tasks fully without seeking confirmation. Your role is to
↪  deliver comprehensive responses in one go. Never ask if the user wants you to continue or show more –
↪  you must provide the complete response.
```

Figure 18: Claude System Prompt.

## D.5 Textual Grid Representations

$$\begin{bmatrix} 0 & 0 & 0 & 1 & 0 & 0 & 0 \\ 0 & 1 & 0 & 0 & 1 & 0 & 1 \\ 0 & 0 & 0 & 1 & 0 & 0 & 0 \\ 0 & 1 & 1 & 1 & 0 & 1 & 0 \\ 0 & 0 & 0 & 0 & 0 & 1 & 0 \\ 1 & 0 & 1 & 0 & 1 & 1 & 0 \\ 1 & 0 & 1 & 0 & 0 & 0 & 1 \end{bmatrix}$$

|   | 0 | 1 | 2 | 3 | 4 | 5 | 6 |
|---|---|---|---|---|---|---|---|
| 0 | · | · | · | — | · | · | · |
| 1 | · | — | · | · | — | · | — |
| 2 | · | · | · | — | · | · | · |
| 3 | · | — | — | — | · | — | · |
| 4 | · | · | · | · | · | — | · |
| 5 | — | · | — | · | — | — | · |
| 6 | — | · | — | · | · | · | — |

(a) 2D array format (ARC-style).          (b) Markdown-style grid with symbols.

Figure 19: Two text representations of the puzzle grid: array (left) and markdown (right).

