# OpenReview forum: "CrossWordBench: Evaluating the Reasoning Capabilities of LLMs and LVLMs with Controllable Puzzle Generation"
_colmweb.org/COLM/2025/Conference — COLM 2025_

### Official Review · Reviewer_wJrT · 2025-05-10

**Rating:** 7
**Confidence:** 4
**Ethics Flag:** 1

**Summary:**

This paper introduces CrossWordBench, a novel benchmark designed to evaluate the multimodal reasoning capabilities of both LLMs and LVLMs through the task of solving crossword puzzles. Unlike existing benchmarks that separately assess either textual reasoning or vision-language understanding, CrossWordBench requires models to integrate semantic constraints from textual clues and structural constraints from crossword grids. The benchmark introduces a controllable puzzle generation framework that supports multiple formats (text and image) and includes both direct and interactive evaluation modes. Experiments on 20+ models show that reasoning-capable LLMs significantly outperform non-reasoning models, while LVLM performance strongly correlates to their ability to parse visual grids. These results highlight current limitations in multimodal reasoning and suggest directions for future evaluations.

**Questions To Authors:**

Could the authors suggest potential training strategies or data augmentation techniques that might improve the performance of open-weight models on this benchmark?

**Reasons To Accept:**

* Novel benchmark design: The paper presents a controllable crossword puzzle generation framework that uniquely combines semantic reasoning from textual clues with structural reasoning from crossword grid constraints. The benchmark supports diverse formats (text and image) and includes both direct and interactive evaluation modes, enabling a broad assessment of model capabilities.
* Thorough evaluation and insightful analysis: The experiments offer a comprehensive evaluation across a wide range of LLMs and LVLMs, revealing clear limitations in current models’ multimodal reasoning abilities and offering insights such as LVLM performance is strongly tied to their ability to parse visual grids.

**Reasons To Reject:**

* Lack of proposed solutions: While the benchmark identifies current limitations of LLMs and LVLMs in multimodal reasoning, the paper does not explore training strategies or data augmentation methods to address these issues, resulting in its limited technical contribution.

---

> ### Author Response · Authors · 2025-06-01
> **Response to Reviewer wJrT**
>
> Thank you very much for your thoughtful comment and for recognizing benchmark design and thorough evaluation. We address your concerns in detail below.
>
> ---
>
> **Q1:** Potential training strategies or data augmentation techniques that might improve the performance of open-weight models on this benchmark?
>
> **A1:** Thank you for your comment. Our primary goal is to introduce a challenging and diverse evaluation suite for LLMs and LVLMs, rather than explore novel learning methods. That said, our benchmark is well-suited to support future training research. Crossword puzzles offer a unique advantage: they have verifiable, constraint-based solutions, much like structured tasks such as Sudoku [1], 2048 [2]. This property makes them ideal for reinforcement learning (RL) and other training paradigms where clear reward signals are critical. We briefly discuss this potential in Section 5.4 of our paper.
>
> Additionally, our controllable puzzle generation framework naturally supports data augmentation, a capability that static online sources cannot offer. Even when using the same set of word–clue pairs, we can generate diverse grid configurations that vary in structure, layout, and clue placement.
>
> **Reference:**
>
> [1] Hrishbh Dalal. Agent 2048: Forging Strategic Gameplay in an AI Through Data, Rewards, and RL. Blog post, March 2024.
>
> [2]  Hrishbh Dalal. Teaching Language Models to Solve Sudoku Through Reinforcement Learning. Blog post, March 2025.

---

> > ### Comment · Reviewer_wJrT · 2025-06-08
> > **Acknowledgment of Rebuttal**
> >
> > After carefully reviewing the authors' responses, I maintain my positive assessment of the work.

---

### Official Review · Reviewer_TLs7 · 2025-05-12

**Rating:** 7
**Confidence:** 4
**Ethics Flag:** 1

**Summary:**

This paper presents an evaluation framework for LLMs in the context of crossword puzzles.  Specifically, the authors present a novel framework for generating small (7x7 and 14x14) English and Chinese crossword puzzles.  Via prompting, they explore the performance of both language-only and language and vision LLMs on dataset of generated crosswords.   The contributions include the methods for generating puzzles, the evaluation metrics and framework and an analysis of the performance of a suite of modern LLMs and LVLMs.  The evaluation includes both reasoning and non-reasoning models.  Results indicate that large reasoning models outperform standard zero-shot methods and that visual language models struggle with this task.

Overall the paper is well-structured and clear with appropriate treatment of the early literature on this task.

**Reasons To Accept:**

This paper represents an interesting take on a task that has fascinated NLP researchers since Littman's work on Proverb. It serves as an interesting benchmark combining traditional word-based reasoning with constraint-based visual processing.

**Reasons To Reject:**

There are 2 limiting possibilities for this benchmark:  1) that the results will be too specific to crosswords and not really tell us much about LLMs ability to employ constraint-based reasoning, or 2) that the word-level results are really just an odd form of Q/A and results on this benchmark don't tell us much more than what we already know with more standard QA benchmarks.

Another minor criticism is the disconnect in the evaluation with earlier work involving actual crosswords. Such puzzles likely involve aspects not captured in the generation scheme used here.  While the issues of data contamination and copyright are real, it would nevertheless to present a small set of results based on real crosswords that have been employed all the way back to the Proverb system.  If only to demonstrate the gap between the kind of data presented in CrossWordBench and "real" crosswords.  Another avenue of extrinsic evaluation would be the NY Times "mini" crossword.

---

> ### Author Response · Authors · 2025-06-01
> **Response to Reviewer TLs7**
>
> Thank you very much for your thoughtful comment and for recognizing our work. We address your concerns in detail below.
>
> ---
>
> **Q1:** That the results will be too specific to crosswords and not really tell us much about LLMs ability to employ constraint-based reasoning
>
> **A1:** Thank you for your concern about generalizability. Crosswords serve as a canonical constraint-satisfaction problem, and we leverage them as a medium to probe LLMs and LVLMs’ ability to perform constraint-based reasoning. We further show that our framework can generate puzzles drawn from various sources (and even adapt existing benchmarks). This diversity ensures that models are not overfitting to a single corpus or puzzle format, but instead must handle diverse constraint patterns. Thus, although we focus on crosswords, the benchmark provides a meaningful proxy for broader constraint-satisfaction capabilities.
>
> ---
>
> **Q2:** The word-level results are really just an odd form of Q/A and results on this benchmark don't tell us much more than what we already know with more standard QA benchmarks
>
> **A2:** Thank you for the comment. We agree that at a surface level, the task of answering individual crossword clues may resemble traditional question answering (QA). In fact, this is the behavior we often observe in non-reasoning models, which tend to treat each clue independently---ignoring the structural dependencies imposed by the grid.
>
> This is precisely why we introduce complementary metrics, such as ICR, which measures whether the model’s predictions are mutually consistent at crossing points---something not captured in standard QA.
> These metrics allow us to move beyond answer correctness and probe whether models can reason holistically under structural constraints.
>
> ---
>
> **Q3: Another minor criticism is the disconnect in the evaluation with earlier work involving actual crosswords.**
>
> **A3:** Thank you for the feedback. Prior work such as LR2Bench [1] has directly used human-created puzzles (e.g., from LA Times and Vulture), and reported model performance using S-Acc, a metric conceptually similar to our Word Coverage Rate (WCR) Compared to human-created puzzles.  As shown in their Appendix (Table 6), our generated 7x7 English puzzles are more difficult than 5x5 human puzzles, which more closely mirror the complexity of a standard NYT mini crossword.
> While we do not include human-created puzzles in this version, our generation process is designed to preserve key structural properties---such as clue distribution, word length, and blocked cell ratio---as reported in Table 1.
>
> We will include the discussion and make it clearer in the final version.

---

> > ### Comment · Reviewer_TLs7 · 2025-06-08
> >
> > Thanks for your responses. With respect to Q1, I'd say that whether this task serves as a useful proxy for constraint-based problem solving remains to be seen.  With respect to Q3, I would look forward to seeing your proposed discussion. Overall, I like this paper and stand by original score of 7.

---

### Official Review · Reviewer_KRbZ · 2025-05-13

**Rating:** 7
**Confidence:** 3
**Ethics Flag:** 1

**Summary:**

The authors propose "CrossWordBench" which is intended to evaluate VLMs and LMs reasoning abilities over crossword puzzles. They produce a crossword generator which can vary difficulty level and other characteristics, sidestepping various issues in prior work using static crossword puzzle sets.

This is another task that reasoning models clearly outperform LMs on, and one that VLMs struggle at slightly compared to LMs.

They collect existing clue, word pairs and can generate text or visual puzzles from this. They evaluate models both in zero-shot and multi-turn settings, in English and Chinese.

They introduce metrics to score partially completed crosswords including word accuracy, character accuracy, and intersection coherence.

In addition to simple zero-shot eval, they evaluate VLMs in an agentic, interactive mode.

**Questions To Authors:**

Elements of your evaluation like intersection accuracy strictly apply to how an LM would solve a task like this---no human will fill in, or even think to fill in, a crossword with intersecting words that don't match with their intersecting letter. This feels like a more "catastrophic" failure that renders their understanding of the puzzle as a whole mistaken, as opposed to word or character-level errors in an otherwise valid puzzle. So, I am at a loss for how to interpret cases where WCR is significantly higher than ICR; is the model just ignoring intersection consistencies and filling in a bunch of words with coherent, but wrong examples? Does the model have a chance to

**I would like to see a discussion of the interplay between these metrics that maybe addresses some of these questions in the CR if accepted.**

**Reasons To Accept:**

Useful and sensible benchmark design. Major pluses include being dynamic, which is novel relative to other crossword benchmarks.

There are a lot of interesting findings in here, including the tendency of VLM-based agents to succeed conditioned on not failing on the first few guesses which could start future analysis.

Fairly comprehensive set of models evaluated.

**Reasons To Reject:**

Some method details are vague in the main text and require too many trips to the Appendix. For example, the description of the interactive mode agent is not given, but can be kind of inferred from reading the example prompts in the appendices. They don't even tell you in the main text that these prompts are in the appendices, though. I'm assuming the implementation is a simple while loop that starts with the initial prompt and then

In the text-only mode, how is the input puzzle formatted? Were ablations on this format performed like the VLM format ablations?

**Please make some of these implementation details a little easier to comprehend in the main text if accepted.**

---

> ### Author Response · Authors · 2025-06-01
> **Response to Reviewer KRbZ**
>
> Thank you very much for your thoughtful comment and for recognizing the value of our benchmark design. We address your concerns in detail below:
>
> ---
>
> **Q1:** Some method details are vague in the main text and require too many trips to the Appendix.
>
> **A1:** Thank you for your feedback. We will revise the main text to include clearer descriptions of key methods and reduce the need to reference supplementary sections in the final version.
>
> ---
>
> **Q2:** In the text-only mode, how is the input puzzle formatted? Were ablations on this format performed like the VLM format ablations?
>
> **A2:** Thank you for the question. In the text-only mode, we format the puzzle grid following the design inspired by ARC-AGI [1]. The grid is represented as a 2D array, where 1 denotes a black (blocked) cell and 0 denotes an empty (unfilled) cell. For example: “empty grid = [[0, 0, 0, 1, 0, 0, 0], [0, 1, 0, 0, 1, 0, 1], [0, 0, 0, 1, 0, 0, 0], [0, 1, 1, 1, 0, 1, 0], [0, 0, 0, 0, 0, 1, 0], [1, 0, 1, 0, 1, 1, 0], [1, 0, 1, 0, 0, 0, 1]]”
>
> Following your suggestion, we also conducted an ablation study using an alternative formatting scheme---a markdown-style grid, where ‘·’ represents an unfilled cell and ‘-’ represents a blocked cell. For example:
> empty grid =
> ```
> | | 0 | 1 | 2 | 3 | 4 | 5 | 6 |
> |---|---|---|---|---|---|---|---|
> |0| · | · | · | - | · | · | · |
> |1| · | - | · | · | - | · | - |
> |2| · | · | · | - | · | · | · |
> |3| · | - | - | - | · | - | · |
> |4| · | · | · | · | · | - | · |
> |5| - | · | - | · | - | - | · |
> |6| - | · | - | · | · | · | - |
> ```
>
> We have attached the results for your reference, in summary, there is no significant difference in performance between the two formats.
>
> ---
> | Model                  | WCR (Array) | WCR (Markdown) |
> |------------------------|---------------|--------------------|
> | GPT-4o-2024-11-20 (Text)      | 0.410         | 0.398              |
> | Claude-Sonnet-3-7 (Text)      | 0.479         | 0.469              |
> | Gemini-2.0-Flash (Text)       | 0.301         | 0.309              |
> | DeepSeek-V3 (Text)            | 0.303         | 0.294              |
> | o3-mini (Text)                | 0.587         | 0.592              |
> ---
>
> **Reference:**
>
> [1] On the measure of intelligence. François Chollet. Arxiv 2019.
>
> ---
>
> **Q3:** I am at a loss for how to interpret cases where WCR is significantly higher than ICR
>
> **A3:** Thank you for the question. We would like to clarify the distinction between WCR and ICR:
>
> * WCR measures the fraction of words in the puzzle that are predicted exactly correctly, comparing each word to the ground truth.
> * ICR, on the other hand, measures whether the model’s own predictions are internally consistent at crossing points---i.e., whether the same letter appears at the intersecting cell from both the across and down words.
>
> These two metrics capture complementary aspects of model performance. For example, consider a case where two words intersect at one letter:
>
> * The ground truth answers are:
>    * Across: ABCD
>    * Down: BBEF (with the intersection at the second letter “B”)
> * A model might predict:
>    * Across: ABCD (correct)
>    * Down: CCEF (incorrect)
>
> In this case:
> * WCR = 0.5 (one correct word out of two)
> * ICR = 0.0 (because the intersection is inconsistent: ‘B’ vs ‘C’)
>
> Such discrepancies often occur with non-reasoning models that treat crossword solving as independent QA tasks, without enforcing structural constraints across answers.
> This is why we include both metrics: WCR reflects answer accuracy, while ICR diagnoses whether a model reasons holistically over the grid structure. We will make this distinction clearer in the final version.

---

> > ### Comment · Reviewer_KRbZ · 2025-06-09
> >
> > Thank you for your response. I'm not sure what went wrong with my response with some cut-off sentences, I must have copied it incorrectly out of Google Docs or something.
> >
> > Thanks for the ablation. I see there's no clear story for which is better, but it's good to have both anyway so readers don't wonder about it as they read.
> >
> > Regarding my question, I do understand what the metrics mean. My point was that *within the rules of crossword puzzles, it shouldn't be "possible" to submit an answer with this type of error*. It is impossible for a human to write two different letters in a crossword square.
> > I understand that part of the point of this study is to evaluate how well LMs can reason around this constraint, but this fundamental type of error could probably be targeted with a more sophisticated prompt system, or having the paper center more on the interactive mode. So, I'm glad you did include the interactive mode experiment in your study.
> >
> > Anyway, I like the paper and continue to think it should be accepted.

---

### Official Review · Reviewer_7ztd · 2025-05-13

**Rating:** 6
**Confidence:** 4
**Ethics Flag:** 1

**Summary:**

This paper introduces a benchmark for evaluating the reasoning capabilities of  Large Vision-Language Models (LVLMs) using crossword puzzles. The benchmark leverages a controllable puzzle generation framework that produces puzzles in text and image formats from various sources (public repositories, dictionaries, adapted benchmarks). The authors evaluate over 20 models using metrics assessing word/letter accuracy and consistency with crossing constraints (ICR). Key findings include: reasoning LLMs significantly outperform non-reasoning LLMs and LVLMs, particularly in leveraging constraints; LVLMs struggle significantly, with performance correlating strongly with grid-parsing accuracy; even puzzles from saturated benchmarks remain challenging. The paper also proposes an interactive evaluation mode and analyzes performance variations based on crossing letters, self-reflection, and test-time scaling.

**Questions To Authors:**

1. A model could achieve high ICR by being consistently wrong across intersections, providing a false sense of constraint adherence accuracy. Is ICR the best suited metric to be chosen here ?

2. The "Agentic Evaluation Setting" (Sec 4.4) lacks detail on error feedback mechanisms and most models fail immediately (ISS mostly 0 or 1). Can the authors provide more details for the same.

3. Could it be that the strong correlation found between LVLM grid-parsing accuracy and puzzle-solving performance (Fig 3, Table 3), along with the Across/Down bias, suggests that LVLM failures are dominated by fundamental limitations in OCR and fine-grained spatial layout understanding, rather than deficits in higher-level multimodal reasoning about constraints ?

**Reasons To Accept:**

1. Focus on Constraint Satisfaction: The emphasis on intersectional constraints (and the ICR metric) highlights a crucial aspect of reasoning that is often under-explored in standard benchmarks – the ability to satisfy multiple, interlocking hard constraints.

2. Controllable and Scalable Benchmark Generation: The idea of a framework that can generate diverse puzzles from multiple sources (public repos, dictionaries, adapted QA) and in different formats is a significant strength if well-executed. This offers flexibility and potential for creating varied and challenging evaluation sets.

3. Evaluation of Both LLMs and LVLMs: Systematically comparing both types of models on a task that can be framed bimodally provides a broader understanding of current model capabilities and their respective weaknesses.

**Reasons To Reject:**

1. Limited Novelty and Contribution: The use of crosswords in NLP is not new. The paper should show its contribution from prior work beyond citing limitations of existing datasets. Crucially, it fails to articulate why this specific task, compared to existing complex multimodal reasoning benchmarks, offers a more significant or fundamentally different evaluation of the claimed "dynamic interplay" between modalities.

2. Opaque Puzzle Generation: The core "controllable puzzle generation framework" is presented as a black box. Critical technical details regarding the algorithm for word placement, constraint management, black square insertion, and difficulty control (beyond grid size) are missing. It would help the readers if the authors could provide them.

3. Lack of Validation: There is no validation demonstrating that the generated puzzles possess comparable quality, difficulty, or structural properties to standard human-created crosswords. Can there be a comparison between the same. This could strengthen this work. Human baselines could also help.

4. Claims about the benefits of crossing constraints (Sec 5.1) are based on observed correlations rather than controlled variation. If authors could do an ablation study regarding isolating the effects of grid size, clue difficulty/ambiguity, or crossing letter density on performance, it would help the reader.

5. The evaluation does not control for or isolate the impact of image resolution, grid clarity, font variations, or underlying OCR capabilities, making it difficult to disentangle visual perception failures from reasoning failures. A control experiment feeding LLMs the OCR'd output from the grid images could assist to isolate modality-specific effects.

6. Assertions that better ICR indicates superior "reasoning capability" are unsubstantiated; it may merely reflect better pattern matching or constraint satisfaction heuristics specific to this grid format.

---

> ### Author Response · Authors · 2025-06-01
> **Response to Reviewer 7ztd (Part 1)**
>
> Thank you very much for your valuable comment. We will incorporate your suggestions in the final version and address your concerns in detail below:
>
> ---
>
> **Q1:** Limited Novelty and Contribution
>
> **A1:** Thank you for the comment. We agree that crosswords have been explored in prior work; however, our contribution goes beyond in several important ways:
> * We introduce a scalable and controllable generation framework that enables variation of grid sizes, word-clue pairs, input modality (texts and images), and evaluation settings (different prefill ratios and interactive mode), which prior datasets do not support.
> * Crucially, crossword facilitates multimodal reasoning under persistent structural constraints. Unlike other multimodal benchmarks---such as chart understanding [1] or math reasoning [2]---crossword solving requires the model to **dynamically and repeatedly refer to the spatial grid structure throughout the problem-solving process**. The model must not only interpret clues but also coordinate multiple answers to maintain consistency within the intersection of the grid.
> * This creates an environment for evaluating the dynamic interplay between vision and language: models must extract and reason over layout, resolve clues, and ensure that answers conform to both semantic and spatial constraints. Such tight coupling between modalities and reasoning steps is rarely demanded in existing benchmarks.
>
> **Reference:**
>
> [1] ChartQA: A Benchmark for Question Answering about Charts with Visual and Logical Reasoning. ACL Findings 2022
>
> [2] MathVista: Evaluating Mathematical Reasoning of Foundation Models in Visual Contexts. ICLR 2024.
>
> ---
>
> **Q2:** Opaque Puzzle Generation
>
> **A2:** Thank you for pointing this out. We will include a more detailed description of our puzzle generation process in our final version and of course release the code. We felt that for the COLM audience the details of the generation process might be of less interest than the implications of the new benchmark for LLM and LVLM research.
>
> In brief, our generation framework uses a heuristic scoring function that encourages desirable properties such as a higher number of horizontal and vertical intersections between words. We also apply post-processing filters, discarding puzzles that fall below a threshold for minimum clue count or exceed a threshold for blocked cells, to ensure quality of the generated puzzles.
>
> While we primarily use grid size to control puzzle difficulty, these heuristics and filters allow us to filter out low quality puzzles. The statistics of the generated puzzles, including clue count, word length, and blocked cell ratios, are reported in Table 1, and serve as a proxy for puzzle complexity and quality
>
> ---
>
> **Q3:** Lack of Validation: There is no validation demonstrating that the generated puzzles possess comparable quality, difficulty, or structural properties to standard human-created crosswords. Can there be a comparison between the same. This could strengthen this work. Human baselines could also help.
>
> **A3:** Thank you for the suggestion, and we will discuss this further in the final version and consider human baselines for the future. Complementary prior work such as LR2Bench [1] has directly used human-created puzzles (e.g., from LA Times and Vulture), and reported model performance using S-Acc, a metric conceptually similar to our Word Coverage Rate (WCR). Compared to human-created puzzles, our benchmark seems to be of comparable difficulty. However, while it is clear from prior work that these tasks are difficult for LVLMs, it is unclear what the source of that difficulty is.
>
> In our current work, we thus focus on automatically generated puzzles to enable scalable, controllable, and diverse evaluation across different subjects and grid sizes. This design allows us to systematically vary puzzle difficulty (via grid size and word-clue pairs selection) and supports a wide range of evaluation strategies—including pre-filled puzzle settings and an interactive mode for step-by-step reasoning. While we do not include human-created puzzles in this version, our generation process is designed to preserve key structural properties---such as clue distribution, word length, and blocked cell ratios---as reported in Table 1.
>
> In addition, our benchmark offers a broader set of evaluation metrics. These provide complementary, fine-grained signals beyond overall accuracy, making the benchmark more differentiated and diagnostic for assessing model capabilities.
>
> **Reference:**
>
> [1] Lr2bench: Evaluating long-chain reflective reasoning capabilities of large language models via constraint satisfaction problems. Arxiv 2025.

---

> > ### Author Response · Authors · 2025-06-01
> > **Response to Reviewer 7ztd (Part 2)**
> >
> > **Q4:** Claims about the benefits of crossing constraints (Sec 5.1) are based on observed correlations rather than controlled variation. If authors could do an ablation study regarding isolating the effects of grid size, clue difficulty/ambiguity, or crossing letter density on performance, it would help the reader.
> >
> > **A4:** Thank you for your suggestion. In Section 5.1, we report results exclusively on 7x7 English puzzles, thereby controlling for grid size. We further observe that 14x14 puzzles do not exhibit the same pattern---likely because 14x14 puzzles are substantially more difficult even for reasoning models---and we include the results below for reference.
> >
> > ---
> > | Grid Size | Range     | Reasoning LLMs  | Nonreasoning LLMs  |
> > |------------------|-----------|---------------------------------|-------------------------------------|
> > | 7x7      | low (1)   | 0.4539          | 0.2922               |
> > | 7x7       | medium (2)| 0.5308         | 0.3230                |
> > | 7x7       | high (3+) | 0.6183         | 0.3259                |
> > | 14x14   | low (1)   | 0.4115           | 0.3469              |
> > | 14x14   | medium (2)| 0.4186          | 0.3514               |
> > | 14x14   | high (3+) | 0.2306          | 0.1438             |
> > ---
> >
> > To isolate crossing-letter density within a constraint grid, we conduct a quasi-ablation on 7x7 puzzles by stratifying puzzles into low, medium, and high density based on crossing letters per word. We have attached the table for reference. In summary, within different density, reasoning LLMs’ WCR performance increases with more number of crossing letters, while non-reasoning LLMs remain roughly constant.
> >
> > ---
> > | Density Category | Range     | Reasoning LLMs  | Nonreasoning LLMs  |
> > |------------------|-----------|---------------------------------|-------------------------------------|
> > | low_density (40 puzzles)      | low (1)   | 0.4341          | 0.2960               |
> > | low_density      | medium (2)| 0.5275           | 0.3355                |
> > | low_density      | high (3+) | 0.6165          | 0.2794                |
> > | medium_density (33 puzzles)   | low (1)   | 0.4271           | 0.2754              |
> > | medium_density   | medium (2)| 0.5477          | 0.3089               |
> > | medium_density   | high (3+) | 0.6029          | 0.4193             |
> > | high_density  (27 puzzles)   | low (1)   | 0.5288       | 0.3097               |
> > | high_density     | medium (2)| 0.5151        | 0.3217              |
> > | high_density     | high (3+) | 0.5928           | 0.3030              |
> > ---
> >
> > We acknowledge that without per-clue difficulty or ambiguity annotations it remains challenging to disentangle any subtle clue-level effects fully, and we will discuss these limitations in the final version.
> >
> > ---
> >
> > **Q5:** A control experiment feeding LLMs the OCR'd output from the grid images could assist to isolate modality-specific effects.
> >
> > **A5:**  Thank you for the suggestion. In our experiments, we find that while OCR models and LVLMs can typically extract clues from images accurately, they struggle to follow instructions and extract a spatially meaningful representation of the empty grid. For example, when applying stepfun-ai/GOT-OCR-2.0-hf to an image containing both the grid and clues, it generates “Across:\n2. Canon product, for short\n4. Defense aid\n6. Certain fraud protector, for short\n7. Abbr. before a founding date\n10. Philosopher\'s study\n12. May honoree\n13. Fraternity letter\nDown:\n1. Like white panthers\n2. [Not my mistake]\n3. Beta dog\'s view\n5. Gridiron abbr.\n8. One of the muskrats in the 1976 hit "Muskrat Love"\n9. Slow-witted\n11. Going rate?: Abbr.\n”. While the clue text is extracted correctly, the output contains no representation of the grid structure. This makes it unsuitable as input for LLMs, as the structural constraints---critical for crossword solving---are lost.
> >
> > On the other hand, the clue content extracted by OCR models and LVLMs is nearly identical to what we explicitly feed to LLMs in the text input setting. In fact, we augment the clue text with explicit positional hints to compensate for the lack of visual spatial information—details that LVLMs might infer from the image but LLMs cannot derive from plain text alone.
> >
> > To further isolate the impact of input modality, we conduct an experiment in Appendix A.6 (Figure 9), where we compare two LVLM input formats:
> > * An image containing both the empty grid and clues.
> > * An image of the empty grid only, with clues provided in text.
> >
> > The results show no significant difference in Word Coverage Rate (WCR) between these formats, indicating that OCR capability is not the only bottleneck. Rather, current LVLMs also struggle with spatial reasoning over the visual grid, even when clue text is provided clearly.  We will add this discussion to the final version.

---

> > > ### Author Response · Authors · 2025-06-01
> > > **Response to Reviewer 7ztd (Part 3)**
> > >
> > > **Q6:** Assertions that better ICR indicates superior "reasoning capability" are unsubstantiated;it may merely reflect better pattern matching or constraint satisfaction heuristics specific to this grid format. A model could achieve high ICR by being consistently wrong across intersections, providing a false sense of constraint adherence accuracy. Is ICR the best suited metric to be chosen here?
> > >
> > > **A6:** Thank you for raising this important point. We agree that ICR does not evaluate the correctness of answers with respect to the ground truth. Rather, it measures a model’s ability to maintain internal consistency across its own predictions at intersecting grid positions.
> > >
> > > ICR is not intended to serve as a standalone indicator of reasoning capability, but instead as a complementary diagnostic metric. It reflects whether a model understands and respects the structural constraints of the crossword format.
> > > We acknowledge the possibility that a model could, in theory, achieve high ICR while producing consistently incorrect answers. However, as shown in Table 2, models with higher ICR typically also achieve higher Word Coverage Rate (WCR). This suggests that models which better respect grid constraints also tend to reason more accurately over the clues---indicating that ICR aligns with reasoning ability in practice, even if it does not directly measure it.
> > >
> > > ---
> > >
> > > **Q7:** The "Agentic Evaluation Setting" (Sec 4.4) lacks detail on error feedback mechanisms and most models fail immediately (ISS mostly 0 or 1). Can the authors provide more details for the same.
> > >
> > > **A7:** Thank you for the question. We agree that the Agentic Evaluation Setting introduced in Section 4.4 is still preliminary. Our goal in this paper is not to present a complete agent-based framework, but rather to demonstrate the potential of our puzzle generation system to support such evaluations in the future.
> > >
> > > In our current setup, the puzzle generator can be exposed as a function, which allows models to receive basic feedback signals, such as:
> > > * Whether a proposed word fits the expected length.
> > > * Whether the intersection characters match existing filled cells.
> > >
> > > However, we do not implement multi-step error correction or re-try mechanisms in this version. As a result, models typically fail early (i.e., low Interactive Success Step (ISS)), often due to an incorrect guess that violates grid constraints---causing the interaction to end.
> > >
> > > Our intent is to provide a foundation for agentic interaction, where a model can iteratively reason, receive structured feedback, and refine its predictions.  We will clarify this in the final version.
> > >
> > > ---
> > >
> > > **Q8:** Could it be that the strong correlation found between LVLM grid-parsing accuracy and puzzle-solving performance (Fig 3, Table 3), along with the Across/Down bias, suggests that LVLM failures are dominated by fundamental limitations in OCR and fine-grained spatial layout understanding, rather than deficits in higher-level multimodal reasoning about constraints?
> > >
> > > **A8:** Thank you for the thoughtful observation. We agree that the strong correlation between LVLM grid-parsing accuracy and puzzle-solving performance (as shown in Figure 3 and Table 3) highlights the limitations of current LVLMs in low-level spatial layout understanding, including OCR and grid parsing.
> > >
> > > In fact, we explicitly discuss this issue in Section 4.3, where we analyze the model's parsing accuracy for Across vs. Down clues and show that LVLMs often misinterpret or overlook spatial relationships within the grid. This suggests that failures stem not only from reasoning challenges but also from fundamental limitations in visual perception and layout understanding.
> > >
> > > To further isolate this factor, we conduct a control experiment in Appendix A.6 (Figure 9) where we compare two LVLM input formats:
> > > * An image containing both the grid and clues.
> > > * An image of the empty grid only, with clues provided in text.
> > >
> > > This setup reduces the dependency on OCR by explicitly supplying the clue content in text form. However, we observe no significant improvement in performance under this condition, suggesting that the key limitation lies in fine-grained spatial understanding, not just text recognition.

---

> ### Comment · Reviewer_7ztd · 2025-06-06
> **Thank you for your rebuttal**
>
> Thank you for your responses.
>
> Please add a detailed version of A2, A3 in the camera ready paper. Please add A4 in the camera ready version of the paper as well. I have increased my score.

---

### Official Review · Reviewer_XYjy · 2025-05-14

**Rating:** 5
**Confidence:** 4
**Ethics Flag:** 1

**Summary:**

This paper presents CrossWordBench, a novel benchmark designed to evaluate the reasoning capabilities of LLMs and LVLMs through crossword puzzles that integrate both semantic (text-based clues) and structural (grid-based) constraints. The authors develop a controllable puzzle generation pipeline that supports both image and text formats, enabling direct and interactive evaluation modes. By evaluating over 20 state-of-the-art models, the study finds that reasoning LLMs outperform non-reasoning models, especially in leveraging crossing-letter constraints, while LVLMs underperform significantly due to weak grid parsing abilities. The benchmark reveals substantial gaps in multimodal reasoning, offering a useful testbed for future research on constrained, verifiable tasks in both LLMs and LVLMs.

**Questions To Authors:**

N/A

**Reasons To Accept:**

* **The dataset is useful and may have potential to serve as a standardized testbed** for evaluating multimodal reasoning under structural constraints, which is currently underexplored in existing benchmarks. By leveraging crossword puzzles that integrate both textual clues and spatial grid constraints, the benchmark opens up new directions for studying verifiable and compositional reasoning in both LLMs and LVLMs.

* **The evaluation is comprehensive, covering a wide range of models** including both proprietary and open-weight LLMs and LVLMs, reasoning and non-reasoning variants, and multilingual settings. The authors also explore different puzzle formats, grid sizes, reasoning strategies, and interactive solving setups, providing a thorough analysis of model performance under varied conditions.

* **The paper is well-written and clearly presented**, with intuitive figures, detailed methodology, and insightful analysis. The empirical findings are clearly tied back to the motivation and demonstrate the practical value of the benchmark.

**Reasons To Reject:**

* **The novelty and scope of the benchmark may be limited** in terms of its ability to generalize beyond crossword-style tasks. While the integration of textual and visual constraints is interesting, the setting may not fully capture the breadth of reasoning challenges faced in real-world applications of LLMs and LVLMs.

* **The findings offer limited new insights** beyond confirming that reasoning models perform better and that LVLMs struggle with visual parsing. Many of the conclusions align with existing understanding, and there is little discussion on deeper failure modes or unexpected behaviors.

* **The paper does not sufficiently explore different training or inference methods**, such as supervised fine-tuning, reinforcement learning tunning, test-time scalling, or more advanced prompting strategies (e.g., multi-step or multi-agent prompting). This limits the utility of the benchmark as a tool for driving methodological innovation.

---

> ### Author Response · Authors · 2025-06-01
> **Response to Reviewer XYjy**
>
> Thank you very much for your valuable comment. We will address your concerns in detail below:
>
> ---
>
> **Q1:** The novelty and scope of the benchmark may be limited
>
> **A1:** Thank you for your feedback. We believe that the novelty and importance of the benchmark is supported by two observations: (1) since small crosswords are puzzles that people can do fairly easily, but LVLMs struggle with, this suggests a weakness in current LVLMs and (2) the reasoning required in our benchmark is quite different from that in most other multimodal reasoning benchmarks.  We expand on point (2) below.
>
> Our benchmark adopts a crossword-style formulation that requires multi-step, multimodal constraint satisfaction. Unlike many existing multimodal reasoning datasets---often framed as visual question-answering (VQA) tasks [1, 2, 3, 4]---our benchmark combines clues with a visual grid, where QA pairs intersect and mutually constrain each other. Solving one entry often necessitates iterative refinement of other answers, introducing a structured reasoning loop absent from typical VQA formulations. Compared to structured reasoning benchmarks for constraint satisfaction problems [5, 6], which are typically text-only, our task supports both multimodal input generation and vision-language evaluation, enabling a more holistic test of LVLMs.
>
> Moreover, the underlying reasoning paradigm generalizes to real-world scenarios. For instance, embodied agents like warehouse robots or autonomous vehicles must coordinate between visual observations and textual instructions under structural constraints.
>
> **Reference:**
>
> [1] ChartQA: A Benchmark for Question Answering about Charts with Visual and Logical Reasoning. ACL Findings 2022
>
> [2] DocVQA: A Dataset for VQA on Document Images. WACV 2021
>
> [3] MathVista: Evaluating Mathematical Reasoning of Foundation Models in Visual Contexts. ICLR 2024.
>
> [4] MMMU: A Massive Multi-discipline Multimodal Understanding and Reasoning Benchmark for Expert AGI. CVPR 2024 Oral
>
> [5] Lr2bench: Evaluating long-chain reflective reasoning capabilities of large language models via constraint satisfaction problems. Arxiv 2025.
>
> [6] ZebraLogic: On the Scaling Limits of LLMs for Logical Reasoning. ICML 2025.
>
> ---
>
> **Q2:** The findings offer limited new insights or discussion on deeper failure modes
>
> **A2:** Thank you for your comment. In our study, we observe:
>
> * OCR orientation failures: LVLMs often mis-extract vertical entries, indicating weak spatial grounding or a bias towards horizontal layouts
> * Ineffective self-reflection: Adding self-reflection mechanism yields no noticeable gains, showing that prompting the model (both reasoning and non-reasoning ones) to critique its own reasoning is insufficient for solving crossword puzzles.
> * Effect of crossing-letter constraints: When a word has more intersections, reasoning LLMs exhibit higher accuracy, whereas non-reasoning LLMs do not benefit—suggesting that non-reasoning LLMs struggle to leverage these structural constraints.
>
> We will also include some case studies in the final version.
>
> ---
>
> **Q3:** The paper does not sufficiently explore different training or inference methods
>
> **A3:** Thank you for your comment. Our primary goal is to introduce a challenging and diverse evaluation suite for LLMs and LVLMs, rather than explore novel learning methods. That said, our benchmark is well-suited to support future training and inference research. In particular, crossword puzzles offer a valuable property: they have verifiable solutions, similar to structured tasks like Sudoku [1] and 2048 [2]. This makes them an excellent candidate for reinforcement learning with verifiable rewards, as we discussed in Sec 5.4 of our paper. Moreover, our controllable generation framework allows for diverse puzzle generation.
>
> Additionally, our interactive mode does represent a novel inference scheme. While the results are preliminary, this provides a good foundation for future work on agentic evaluation, where models make decisions over multiple steps and can receive feedback via function calls to the puzzle generator.
>
> We also explore a self-reflection prompting strategy in Section 5.2. While this particular method did not lead to significant improvements---either for reasoning LLMs or non-reasoning LVLMs---it demonstrates how our benchmark can be used to evaluate the effectiveness of various inference-time scaling strategies.
>
> **Reference:**
>
> [1] Hrishbh Dalal. Agent 2048: Forging Strategic Gameplay in an AI Through Data, Rewards, and RL. Blog post, March 2024.
>
> [2] Hrishbh Dalal. Teaching Language Models to Solve Sudoku Through Reinforcement Learning. Blog post, March 2025.

---

> > ### Comment · Reviewer_XYjy · 2025-06-10
> > **Response to author's rebuttal**
> >
> > Thanks for providing the rebuttal for my review. It solve my concerns to a certain degree. However, my concerns on different training or inference methods still remain and I will keep my scores.

---

> ### Author Response · Authors · 2025-06-07
> **Follow-up discussion (3 days left before the discussion period ends)**
>
> Thank you so much for your dedicated review of our paper. We recognize the significant time and effort involved in your review, and we greatly appreciate it. With only 3 days remaining before the conclusion of the discussion phase, we wish to extend a respectful request for your feedback about our responses. If there are any questions or unresolved issues, we are eager to provide further clarification or make necessary revisions. Thank you!

---

### Official Review · Reviewer_kmED · 2025-05-21

**Rating:** 6
**Confidence:** 4
**Ethics Flag:** 1

**Summary:**

The paper proposes a new benchmark for evaluating multimodal and text based reasoning and non-reasoning models on crossword tasks. The authors dont rely on public benchmarks but rather on novel curated crosswords through puzzles, dictionaries, etc. The authors also compare the performance on different models and find interesting insights like (1) vision models are worse than language models; (2) reasoning models are always better.

**Questions To Authors:**

- Why was the text input not given to vision models. Otherwise this makes it an OCR task.
- Why not GPT-4o non vision model?
- In multi shot setting, which clue does the model choose to answer? the first one or any one it chooses?
- In multi shot, if the model runs into an error where a prediction cannot be added to the grid, what happens? does the turn end?
- Were any pairs filtered? for non obvious dictionary words for example?

**Reasons To Accept:**

- The crossword problem is interesting and a good task to evaluate the models on reasoning and visual understanding.
- The evaluation is thorough and offers interesting insights.
- The crossword construction can be extended to specific domains and more sophisticated benchmarking.
- The proposed metric (performance on intersection) is interesting.

**Reasons To Reject:**

- The vision models are not given the text input also which is a big threat to validity for the hypothesis (vision models are worse). It would be more interesting to see if the vision models are given additional input and how that helps.
- The 1 shot setup seems very restrictive and not sure if the evaluation in 1 shot is fair.
- Was any qualitative study done on the crosswords? stats like number of instructions will validate the quality of the data.
- All vision models are worse than their text only counterparts, which indicates that the problem is not equivalent for vision and non visison.

---

> ### Author Response · Authors · 2025-06-01
> **Response to Reviewer kmED (Part 1)**
>
> Thank you very much for your valuable comment. We will address your concerns in detail below:
>
> ---
>
> **Q1:** The vision models are not given the text input also which is a big threat to validity for the hypothesis (vision models are worse). It would be more interesting to see if the vision models are given additional input and how that helps.
>
> **A1:** Thank you for raising this concern regarding the input modality. To address this, in the submission we conduct experiments where LVLMs are provided with different input formats. Specifically, Table 2 of our main paper includes results from several models (GPT-4o-2024-11-20, Claude-3-7-Sonnet, Gemini-2.0-Flash) under **both text-only and image-only** input settings. Across these models, performance is consistently lower when provided only with image inputs compared to equivalent textual inputs.
>
> Additionally, in Appendix Section A.6, we present an experiment (please see Fig. 9) comparing two input formats for LVLMs:
> - one where the entire puzzle (clues and grid) is rendered as a single image and fed to the model
> - another where the grid is given as an image, but the clues are provided in text.
>
> The results show **no substantial difference in Word Coverage Rate (WCR) between these two configurations**. This suggests that the bottleneck is not merely in the input format, but rather in LVLMs' overall ability to reason over visual input---even when augmented with textural clues.
>
> ---
>
> **Q2:** The 1 shot setup seems very restrictive and not sure if the evaluation in 1 shot is fair.
>
> **A2:** Thank you for your comment. Our main evaluation is conducted in a zero-shot setting, as explained in Section 4.1 and Figures 11 and 12 in the Appendix. Models are not given any in-context examples or demonstrations during inference. All models are evaluated under the same zero-shot setting to ensure fairness and comparability.
>
> We agree that solving an entire crossword puzzle from scratch is a challenging task, even for the small puzzles, and performance could be higher in a k-shot setting. As an alternative to doing this, we did explore ways to reduce task complexity. In particular, leveraging our controllable generation framework, we are able to systematically adjust the difficulty of the task by controlling the pre-fill ratio.
>
> Specifically, we consider a pre-fill ratio of 0.5, meaning 50% of the grid cells are revealed to the model, while ensuring that no complete word is revealed.
>
> We have pasted the results below for your reference. Models are evaluated under two settings (0% and 50% pre-fill) on the English 7x7 puzzles. In both cases, we report WCR.
>
> ---
> | Model                  | WCR (0% Prefill) | WCR (50% Prefill) |
> |------------------------|---------------|--------------------|
> | GPT-4o-2024-11-20 (Text)     | 0.410         | 0.483              |
> | GPT-4o-2024-11-20 (Img)      | 0.348         | 0.383              |
> | Claude-Sonnet-3-7 (Text)     | 0.482         | 0.604              |
> | Claude-Sonnet-3-7 (Img)      | 0.479         | 0.568              |
> | Gemini-2.0-Flash (Text)      | 0.301         | 0.362              |
> | Gemini-2.0-Flash (Img)       | 0.277         | 0.346              |
> | o3-mini (Text)               | 0.587         | 0.864              |
> ---
> As shown in the table, models consistently achieve better performance when provided with more context via pre-filled cells.
> We will include these results in the final version.
>
> ---
>
> **Q3:** Was any qualitative study done on the crosswords to validate the quality of the data?
>
> **A3:** Thank you for your question. In lieu of a formal qualitative study, we will include several examples of generated puzzles in the appendix of the final version.
>
> In Table 1 of the main paper, we did provide detailed statistics of the puzzles across different subjects and grid sizes, including:
> - Average number of words per puzzle.
> - Average word length.
> - Average number of blocked cells.
>
> These metrics can serve as useful proxies for the structural complexity of the puzzles. Additionally, to ensure the diversity and richness of the dataset, we explicitly enforce that no clue is repeated within the same subject across different grid sizes, or even across subjects.

---

> > ### Author Response · Authors · 2025-06-01
> > **Response to Reviewer kmED (Part 2)**
> >
> > **Q4:** All vision models are worse than their text only counterparts, which indicates that the problem is not equivalent for vision and non visison.
> >
> > **A4:** Thank you for the observation and concerns. The only difference between the two setups is the input modality: LLMs receive a textual representation of the grid and clues, whereas LVLMs receive image inputs, either a single image containing both grid and clues or a visual grid paired with separate text clues. We do not observe significant performance difference between the two input configurations for LVLMs (as shown in Appendix Fig. 9). In our experiments, LVLMs accurately extract clue text from images. In other words, the crossword logic and clue content are identical in both settings. Therefore, the performance gap does not stem from differences in the puzzle itself but arise from how well the LVLMs interpret the visual grid.
> >
> > ---
> >
> > **Q5:** Why was the text input not given to vision models. Otherwise this makes it an OCR task. Why not GPT-4o non vision model?
> >
> > **A5:** Please refer to our answer for Q1 (briefly, we do include experiments with text input, and we will clarify/highlight these in the final version).
> >
> > ---
> >
> > **Q6:** In multi shot setting, which clue does the model choose to answer? the first one or any one it chooses?
> >
> > **A6:** Thank you for the question. We think you are referring to the interactive mode described in our paper. In the interactive setting, we do not constrain the model to answer a specific clue. Instead, the model is prompted to choose any clue it wishes to attempt based on the current puzzle state. The exact prompts are shown in Appendix Figures 13 and 14. We will clarify this in the final version.
> >
> > ---
> >
> > **Q7:** In multi shot, if the model runs into an error where a prediction cannot be added to the grid, what happens? does the turn end?
> >
> > **A7:** Thank you for the concern. Yes, in the interactive mode, when the model makes a prediction that cannot be added to the grid, the turn is considered to have failed, and the interactive session ends at that point. We will clarify this in the final version.
> >
> > To quantify performance in this setting, we introduce a metric called Interactive Success Step (ISS), which measures the number of correct words the model successfully fills in before encountering its first error. A detailed discussion of this metric and the interactive mode can be found in Section 4.4 of the main paper.
> >
> > ---
> >
> > **Q8:** Were any pairs filtered? for non obvious dictionary words for example?
> >
> > **A8:** Thank you for the question. We will include a detailed description of our filtering procedures in the Appendix of the final version.
> >
> > For your reference, here is a summary of the filtering steps we applied to ensure the quality and clarity of word-clue pairs across all puzzle types:
> > * For the English and Chinese cryptic puzzles, we first perform clue-level deduplication using cosine similarity to remove redundant or near-duplicate clues. We also filter out context-specific clues (e.g., “1-Across”) and exclude answers that are single letters or pure numbers, as they can be ambiguous or uninformative.
> > * For the English Simple puzzles, which consist of dictionary-style word-clue pairs, we filter out any words that do not have a valid definition in NLTK WordNet.
> > * For CommonSenseQA-based puzzles, we restrict answers to single words and use the associated multiple-choice questions as clues.
> >
> > These filtering steps help maintain the quality, clarity and linguistic relevance of the word-clue pairs used for puzzle generation.

---

> ### Comment · Reviewer_kmED · 2025-06-02
> **Thanks for the rebuttal**
>
> Firs of all I would like to thank the authors for their response. It helps me understand the work a bit better and clarifies my questions.
>
> Some thoughts:
> - The text input study from the appendix should be moved to the main paper. Comparing different input modalities in the same table can be confusing and a threat to validity.
> - The 0-shot setting is restrictive. The prefill results are more indicative but the k-shot analysis should be expanded.
> - The qualitative analysis will help.
> - The interactive mode error handling should be discussed more.

---

> > ### Author Response · Authors · 2025-06-05
> > **Thank you for the feedback**
> >
> > Thank you for the constructive suggestions. We will incorporate your feedback into the final version of the paper and include more discussions on different input modalities, qualitative study, and interactive mode error. Regarding the k-shot setting, we choose not to adopt it in our main evaluation for the following reasons:
> > * Puzzle-level reasoning: We treat each puzzle as a whole rather than as a set of independent QA pairs. This allows us to preserve the structural constraints inherent in crossword solving. Providing a full puzzle solution as a demonstration example would be lengthy and potentially distracting for the model. Moreover, since we employ a model to parse the free-form responses, we do not require demonstrations to guide the model's output format.
> > * Model diversity: Our evaluation includes both reasoning and non-reasoning models. Prior work has observed that CoT in few-shot may hurt the performance of reasoning models [1,2,3].
> > * Modality considerations: For image-based models, few-shot prompting can be distracting as we need to incorporate multiple images into the prompt.
> >
> > That said, following your suggestion, we conduct an additional experiment using 1-shot text-based prompting for non-reasoning LLMs. Specifically, we use a completed puzzle from DeepSeek-R1 on the English_Simple subset as the demonstration. The results are included below for your reference.
> >
> > | Model                  | WCR (0-Shot) | WCR (1-Shot) |
> > |------------------------|---------------|--------------------|
> > | GPT-4o-2024-11-20 (Text)      | 0.410         | 0.394              |
> > | Claude-Sonnet-3-7 (Text)      | 0.479         | 0.519              |
> > | Gemini-2.0-Flash (Text)       | 0.301         | 0.313              |
> > | Llama-3.3-70B-Instruct (Text)       | 0.303         | 0.269              |
> > | DeepSeek-V3 (Text)       | 0.303         | 0.307              |
> >
> > Overall, we find that the performance difference compared to the 0-shot setting is not significant. We will include this analysis in the revised version of the paper.
> >
> > We hope this addresses your concerns and thank you once again for your suggestions!
> >
> > ---
> > **Reference:**
> >
> > [1] DeepSeek-R1: Incentivizing Reasoning Capability in LLMs via Reinforcement Learning
> >
> > [2] Prompt Engineering with Reasoning Models, Blogpost
> >
> > [3] Prompt Engineering for OpenAI’s O1 and O3-mini Reasoning Models, Blogpost

---

### Decision · Program_Chairs · 2025-07-08

**Decision:**

Accept

**Comment:**

This paper proposes a a novel benchmark evaluating LLMs and LVLMs on constraint-based reasoning using dynamically generated crossword puzzles. It is a fun paper, testing another type of reasoning capabilities of foundation models. The authors conduct a thorough evaluation across 20+ models and introduce metrics capturing both accuracy and structural consistency. While some concerns about generalizability and missing methodological details were raised, the authors provided detailed and convincing responses. The benchmark offers a unique testbed for multimodal reasoning and agentic evaluation, and is likely to stimulate further research. The authors should incorporate their responses and address reviewers' comment in the final cam-ready version of this paper.  I recommend acceptance.